# Attosecond control and measurement of chiral photoionization dynamics

Meng Han[1,2,5 ✉], Jia-Bao Ji[1,5], Alexander Blech[3], R. Esteban Goetz[4], Corbin Allison[2], Loren Greenman[2], Christiane P. Koch[3] & Hans Jakob Wörner[1 ✉]

Many chirality-sensitive light–matter interactions are governed by chiral electron dynamics. Therefore, the development of advanced technologies making use of chiral phenomena would critically benefit from measuring and controlling chiral electron dynamics on their natural attosecond timescales. Such endeavours have so far been hampered by the lack of characterized circularly polarized attosecond pulses, an obstacle that has recently been overcome[1,2]. Here we introduce chiroptical spectroscopy with attosecond pulses and demonstrate attosecond coherent control over photoelectron circular dichroism (PECD)[3,4], as well as the measurement of chiral asymmetries in the forward–backward and angle-resolved photoionization delays of chiral molecules. We show that co-rotating attosecond and near-infrared (IR) pulses can nearly double the PECD and even change its sign compared with single-photon ionization. We demonstrate that chiral photoionization delays depend on both polar and azimuthal angles of photoemission in the light-propagation frame, requiring 3D momentum resolution. We measure forward–backward chiral-sensitive delays of up to 60 as and polar-angle-resolved photoionization delays of up to 240 as, which include an asymmetry of about 60 as originating from chirality in the continuum–continuum transitions. Attosecond chiroptical spectroscopy opens the door to quantitatively understanding and controlling the dynamics of chiral molecules on the electronic timescale.

Chirality is traditionally viewed as a structural property of matter. The spatial arrangement of atoms in molecules and materials indeed defines their handedness. In a simplified view, structural chirality is often used to explain chiral recognition. However, recent research indicates that structural chirality is insufficient to fully understand chiral phenomena. In particular, growing evidence points to a functional role of chiral electron dynamics in enabling and mediating chiral interactions at the electronic level[5]. Such dynamics are also thought to play a role in advanced applications, including spintronics[6], unidirectional molecular machines[7] and chiral-sensitive biosensing[8]. Furthermore, a wide range of chiroptical techniques, such as electronic circular dichroism[9] (CD), PECD[10,11] and chiral high-harmonic generation[12,13], are now understood to examine the underlying chiral dynamics of electrons rather than merely structural asymmetries. These developments motivate a broader perspective on chirality that encompasses not only static structure but also the dynamic behaviour of electrons in chiral systems.

In spite of the fundamental importance of chiral electron dynamics, spectroscopy based on attosecond pulses has so far been lacking the crucial capability of chiral discrimination. The key obstacle for expanding attosecond science into the realms of chiral molecules and materials has been the lack of circularly polarized attosecond light pulses. As a consequence, all pioneering experiments in this field have relied on femtosecond pulses, for example, refs. 12–21. Notably, strong-field ionization with intense two-colour femtosecond laser pulses has been used to measure phase shifts[17] and control the asymmetry of photoelectron angular distributions (PADs) in the strong-field ionization of chiral molecules[20,22]. Recently, some of the present authors have developed attosecond metrology in circular polarization by introducing a plug-in apparatus for the generation and demonstration of a general methodology for the complete characterization of circularly polarized attosecond pulses[1]. This new capability has recently been applied to study the photoionization dynamics of atoms[2,23].

Here we introduce attosecond chiroptical spectroscopy by reporting the first application of circularly polarized attosecond pulses to chiral molecules. This development, combined with momentum-vector-resolved electron-ion-coincidence spectroscopy, allows us to measure and control chiral electron dynamics on their natural attosecond timescale. Specifically, our study demonstrates attosecond coherent control over PECD based on the constructive or destructive interference between pairs of well-defined photoionization pathways. It also reveals the characteristic dependence of chiral photoionization delays on both the polar and the azimuthal angles of photoemission in the light-propagation frame.

The experimental set-up is illustrated in Fig. 1a, with details given in Methods. The circularly polarized extreme-ultraviolet (XUV) attosecond pulse train (APT) of a single, common helicity was produced

[1]Laboratorium für Physikalische Chemie, ETH Zürich, Zürich, Switzerland. [2]J. R. Macdonald Laboratory, Department of Physics, Kansas State University, Manhattan, KS, USA. [3]Freie Universität Berlin, Fachbereich Physik & Dahlem Center for Complex Quantum Systems, Berlin, Germany. [4]Department of Physics, University of Connecticut, Storrs, CT, USA. [5]These authors contributed equally: Meng Han, Jia-Bao Ji. ✉e-mail: meng9@ksu.edu; hwoerner@ethz.ch

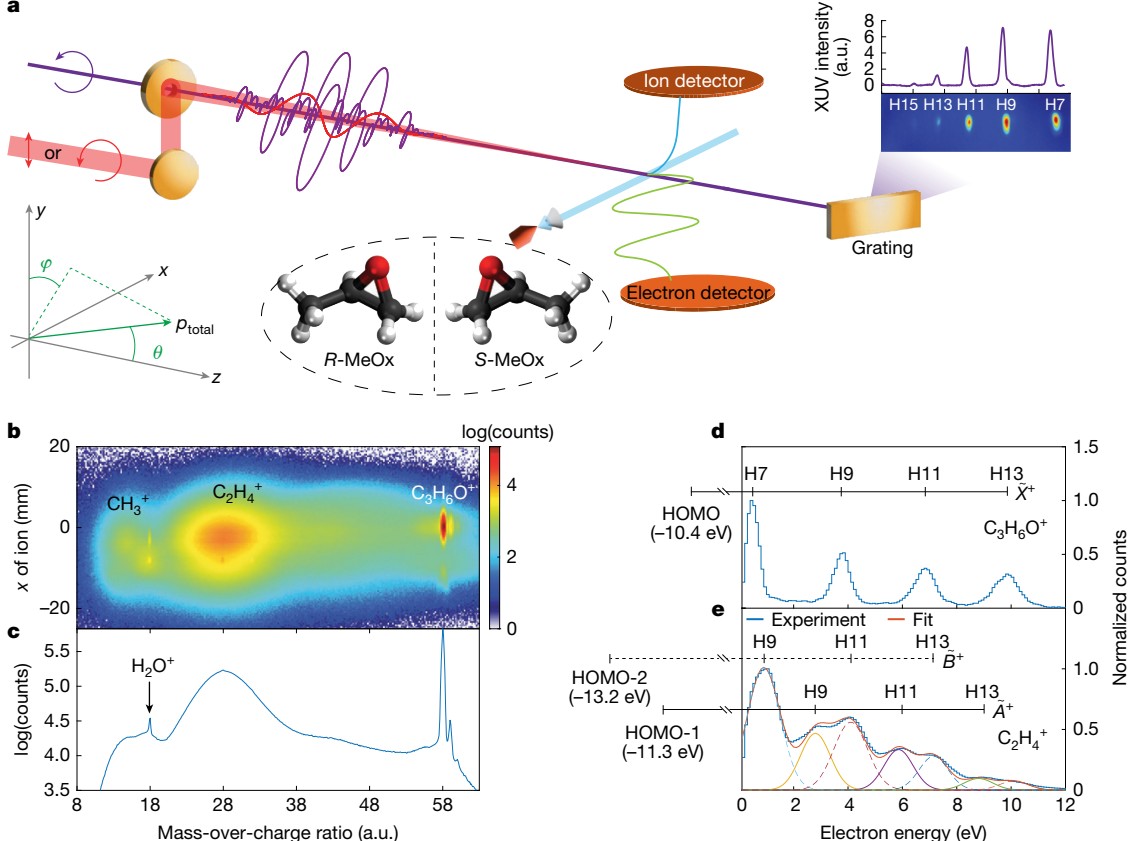

**Fig. 1 | Attosecond chiroptical coincidence spectroscopy. a**, Experimental set-up. The ionic fragments and emitted photoelectrons are measured in coincidence using COLTRIMS. **b**, Measured 2D ionic spectrum as a function of the mass-over-charge ratio and the position on the ion detector along the molecular-beam direction in the presence of the XUV field only. **c**, The projection of the distribution in **b** on the mass-over-charge axis. Note that the counts are shown on a logarithmic scale. The two small peaks above $m/q = 58$ correspond to $^{13}C$ isotopomers of the parent ion. **d,e**, Measured photoelectron spectra in coincidence with the intact parent ion (**d**) and the $C_2H_4^+$ fragment ion (**e**). In **d**, the observation of a series of discrete equidistant peaks confirms that photoionization to the $\widetilde{X}^+$ ground state of the cation leaves the molecule intact. In **e**, the experimental data are fitted by two sets of Gaussian peaks corresponding to the $\widetilde{A}^+$ and $\widetilde{B}^+$ ionic states, respectively. The ionization energies of the relevant states are taken from the ref. 26. a.u., arbitrary units.

by the non-collinear HHG scheme using a compact plug-in apparatus[1]. A weak linearly or circularly polarized (co-rotating) near-IR (800 nm) laser pulse was spatio-temporally overlapped with the XUV APT with attosecond temporal stability to induce XUV–IR two-photon transitions using the reconstruction of attosecond beating by interference of two-photon transitions (RABBIT) approach. In this study, we define the light-propagation direction to be the $z$ axis and the light polarization to be in the $x$–$y$ plane, so that the CD manifests itself in the angular distribution with respect to the angle $\theta = \arccos(p_z/p_{total})$, in which $p_{total}$ and $p_z$ are the total momentum of the electron and its $z$ component, respectively. The azimuthal angle in the polarization plane $\phi = \operatorname{atan}(p_x/p_y)$ was integrated over from −15° to +15°. Enantiomerically pure samples of methyloxirane (MeOx, also known as propylene oxide, $C_3H_6O$, ≥99.5% ee) were delivered into a cold-target recoil-ion-momentum spectroscopy (COLTRIMS) set-up[24,25] by supersonic expansion through the nozzle with an aperture of 30 μm. The 3D momenta of photoelectrons and ionic fragments were measured in coincidence in the ultrahigh-vacuum chamber (about $10^{-10}$ mbar).

Figure 1b shows the ionic fragment distribution as a function of mass-over-charge ratio ($m/q$) and detector position and Fig. 1c presents the projection onto the $m/q$ axis, both recorded using attosecond XUV pulses only. After photoionization by the attosecond pulses comprising harmonic 7 (H7) to harmonic 13 (H13), we observe the undissociated parent ion ($C_3H_6O^+$) and two broader distributions assigned to $CH_3^+$ and $C_2H_4^+$ fragments, in agreement with previous synchrotron studies[26]. The photoelectron spectrum identifies the electronic state of the ion

that was accessed in the ionization step, whereas the ionic species reveal whether and how a given electronic state of the ion dissociates. Figure 1d,e shows the photoelectron spectra measured in coincidence with the parent ion and $C_2H_4^+$, respectively. The former reflects photoionization from the highest occupied molecular orbital (HOMO) by H7–H13, which leaves the cation intact. The photoelectron spectrum measured in coincidence with $C_2H_4^+$ shows a more complex distribution, which contains contributions from the $\widetilde{A}^+$ and $\widetilde{B}^+$ ionic states. In this study, we focus on the parent-ion channel.

Figure 2 shows the measured PECD and its coherent control by changing the XUV–IR delay with attosecond precision for the parent-ion channel. The energy ($E_k$)-resolved and angle ($\theta$)-resolved PECD distribution is defined by $2(I_S(E_k, \theta) - I_R(E_k, \theta))/(I_S(E_k, \theta) + I_R(E_k, \theta))$, in which $I_{S/R}(E_k, \theta)$ is the PAD of the $S/R$ enantiomer. The chiral nature of PECD has also been verified by switching the XUV helicity. Figure 2a,b shows the measured PECD distributions obtained from the XUV field only and the delay-averaged result in the XUV + IR two-colour fields, respectively. For the XUV-only PECD distribution, the four dipole-shaped concentric rings correspond to the four main peaks shown in Fig. 1d. After introducing an IR field (Fig. 2b), sidebands (SBs) appear between the main peaks and inherit the PECD sign of the neighbouring main peaks but with some angular modulations, a consequence of the fact that extra partial waves are involved in the two-photon-ionization process (see the 'Theoretical methods' section for the details). The asymmetry parameters are obtained by fitting trigonometric functions to the angle-resolved PECD, with the results shown in Extended Data Fig. 1.

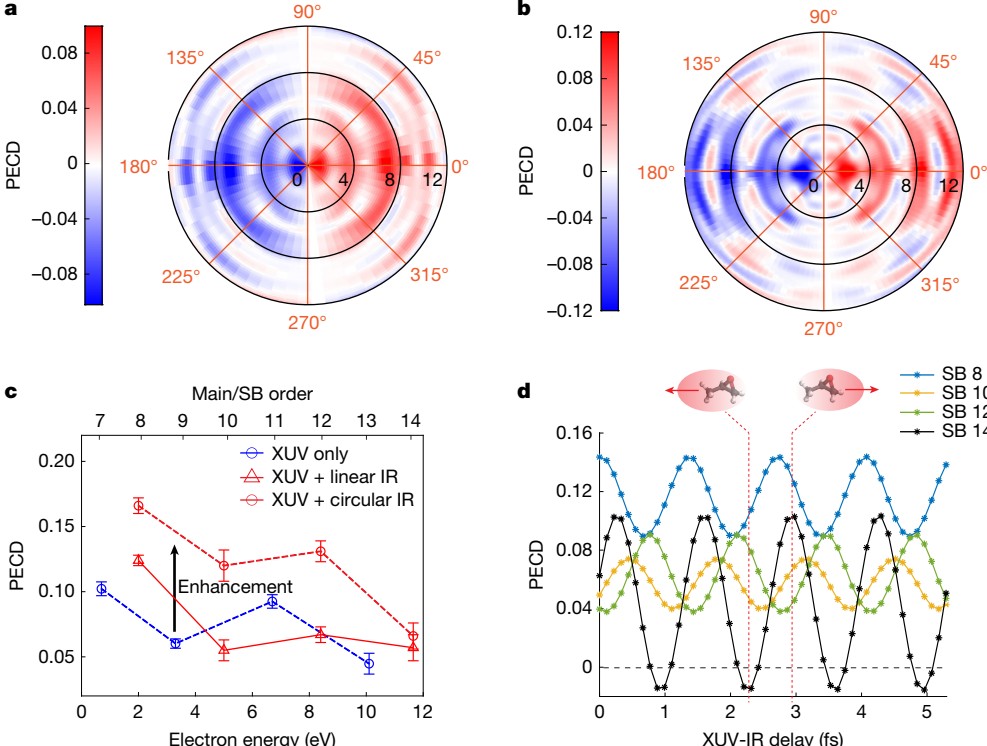

**Fig. 2 | Two-colour enhancement and attosecond coherent control over PECD detected in coincidence with the parent ion. a,b**, Measured angle-resolved and energy-resolved PECD distributions in the XUV-only case and the XUV + IR case with linearly polarized IR field, respectively. **c**, Comparison of PECD values between XUV-only photoionization and the (delay-averaged) XUV + IR two-colour photoionization, including both linear and co-rotating circular polarizations of the IR field. **d**, Measured PECD values at SBs 8, 10, 12 and 14 as a function of the XUV-IR delay for a linearly polarized IR field. The photoelectron signals have been integrated over a window of width 0.5 eV centred at the peaks of the main bands and SBs. The insets above panel **d** illustrate the PECD reversal observed in SB 14.

Figure 2c compares the extracted PECD values in the XUV-only and XUV + IR cases. In the XUV-only case, the highest PECD amounts to about 10%. In the presence of a linearly polarized IR field, the PECDs at the SB positions are enhanced by up to 5% compared with the average PECD of the neighbouring main bands. In the presence of a circularly polarized IR field, the SB PECDs are increased much more, which is in line with predictions from ref. 4 and simulations performed for MeOx. Typically, the PECD values are doubled compared with the average of their neighbouring main bands, as indicated by the black arrow in Fig. 2c. In the XUV + IR field, the PECD can also be actively controlled. Figure 2d shows the delay-resolved PECD in the presence of a linearly polarized IR field. The PECDs at all SB positions are modulated with a period of 1.33 fs, with a maximal modulation depth at SB 14, at which the PECD even changes sign. The phase shifts between the PECD oscillations of different SBs are dominated by the attochirp of the XUV APT. The delay-dependent PECDs in the case of a circularly polarized IR field is shown in Extended Data Fig. 2.

We now discuss the measurement of chiral asymmetries in molecular photoionization delays, starting with the case of a linearly polarized IR field. As in the case of PECD, this approach isolates the chiral signatures of the one-photon XUV transition and eliminates possible chiral contributions from continuum–continuum transitions driven by the IR field[4]. Figure 3a,c shows the energy-resolved RABBIT traces obtained by integrating over the photoelectron emission angle $\theta$ from 0° to 90° (forward-emitted photoelectrons) or from 90° to 180° (backward-emitted photoelectrons). The phases of the photoelectron-yield oscillations at the SB positions have contributions from both the attochirp and the molecular photoionization delays. By evaluating the phase difference between forward-emitted and backward-emitted photoelectrons, the influence of the attochirp is

cancelled, isolating the chiral asymmetry of the molecular photoionization delays. Figure 3d compares the yield oscillations of forward and backward photoelectrons for different SBs, after removal of the slowly varying background by Fourier transformation (see Methods for details). The forward photoelectrons are temporally behind the backward photoelectrons in the case of $R$-MeOx, that is, the SB maxima occur at larger XUV-IR delays, an effect that is most prominent for SB 8.

The same measurements performed on the other enantiomer ($S$-MeOx; Extended Data Fig. 3) showed opposite delays, demonstrating their chiral nature. Figure 3f shows the measured photoionization time delays for both enantiomers. We find that the forward–backward time delays, expressing the temporal separation of the forward-emitted and backward-emitted photoelectron wave packets, amounts to about 60 as at SB 8 and decreases with electron kinetic energy, which reflects the decreasing sensitivity of the photoelectron wave packet to the chirality of the molecular potential, as in the case of PECD amplitudes. The calculated time delays are slightly smaller than the measured ones but lie within the error bars of the measurement. The chiral nature of these forward–backward time delays is confirmed by the opposite signs of the time delays obtained for the two enantiomers in both experiment and theory.

The molecular photoionization dynamics are simulated by adapting the theoretical framework of ref. 4, which suggested making use of RABBIT to realize the coherent control of chiral signatures in the PAD. The simulations are performed in the frozen-core static-exchange and electric-dipole approximations, following refs. 3,4,27 with the modifications described in Methods to remedy the effect of artificial resonances caused by the discretization of the photoelectron continuum and to account for the experimental conditions.

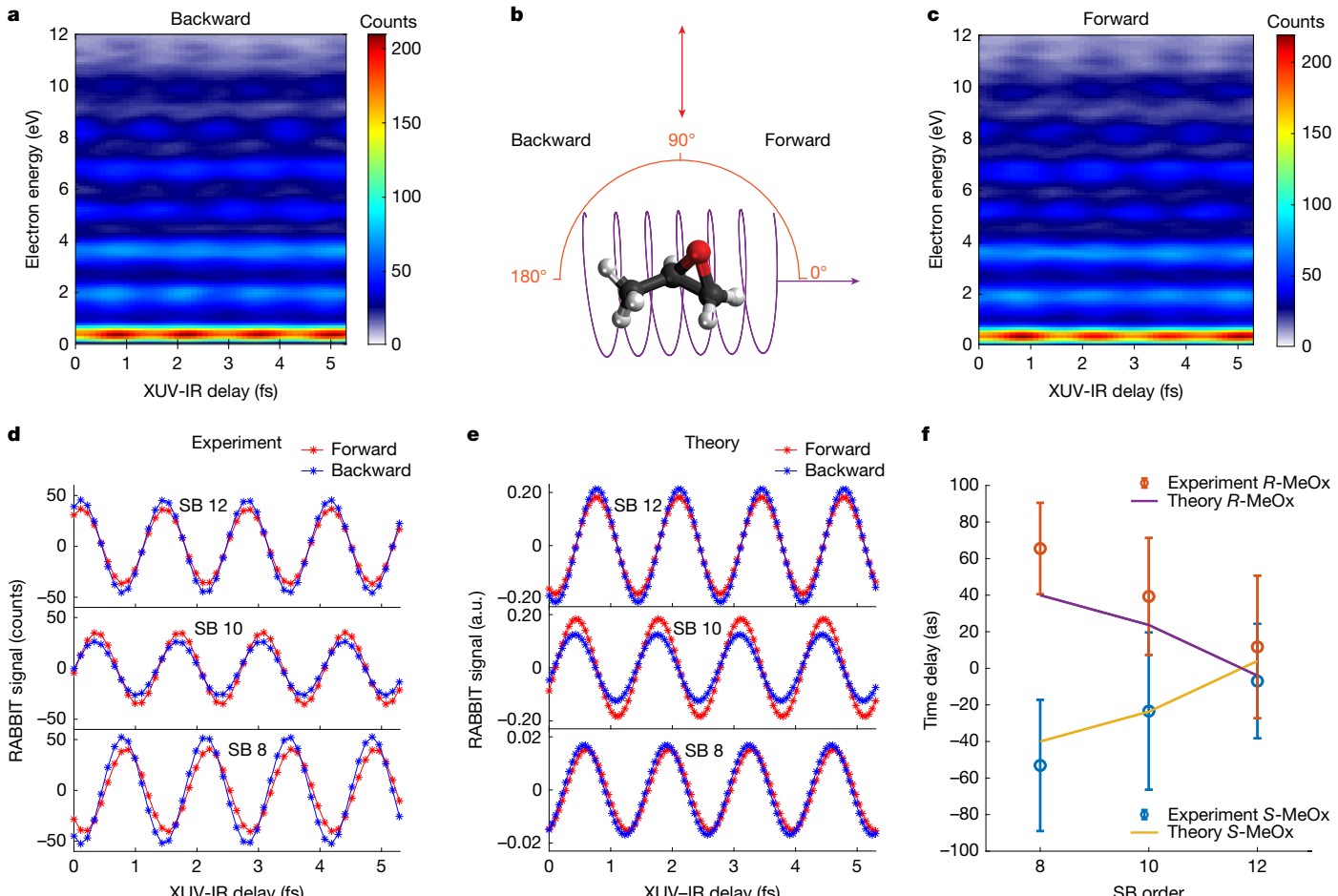

**Fig. 3 | Chiral asymmetries in photoionization time delays.** Data were acquired with a linearly polarized IR along the $y$ direction and the photoemission angle $\phi$ was integrated over from −15° to +15°. **a,c**, RABBIT traces of backward electrons ($\theta$ is integrated over from 90° to 180°) and forward electrons ($\theta$ is integrated over from 0 to 90°), respectively, as shown in panel **b. d,e**, Measured and simulated energy-integrated RABBIT signals for SBs 8, 10 and 12, for which the constant background signals have been removed using Fourier transformation.

The electron-energy integration width is 0.5 eV centred at the peak positions. **f**, Experimental and theoretical forward–backward photoionization time delay as a function of photoelectron energy for both enantiomers of MeOx. Note that the experimental results in panels **a**, **c**, **d** and the theoretical results in panel **e** correspond to $R$-MeOx and the data for $S$-MeOx are shown in Extended Data Fig. 3. a.u., arbitrary units.

The original theory proposal discussed photoelectron interferometry for the model chiral molecule CHBrClF (ref. 4). Compared with CHBrClF, MeOx shows a more isotropic nature, as can be seen from the smaller number of angular basis functions needed to represent the molecular wavefunctions for MeOx compared with CHBrClF (see Methods). This suggests that the photoelectrons of CHBrClF experience higher anisotropy in the molecular potential. To what extent does this influence the robustness of the control scheme? Simulations for CHBrClF in ref. 4 showed that the maximum PECD signal could be enhanced by a factor of five by optimizing the XUV-IR delay, whereas in the simulations for MeOx, optimizing the delay can increase the PECD by about a factor of two, which is a similar enhancement as the delay-averaged experimental results presented in Fig. 2. This may indicate that the presented control scheme could benefit from a less isotropic nature of the molecule. A similar argument can be made when analysing the dependence of the chiral signatures on the photoelectron energy. The PECD as well as the forward–backward time delays decrease with the SB order as shown in Figs. 2c and 3f. Nevertheless, introducing the circularly polarized IR pulse increases the PECD by about 50% for the highest SB (SB 14) compared with the PECD signal of the closest harmonic (H13) in the XUV-only case. The calculation of chiral photoionization delays has been reported in a recent publication[28].

In addition to the chiral forward–backward time delays, the 3D momentum resolution of our experiment allows us to obtain the photoionization time delays with angular resolution. Momentum-vector resolution is a prerequisite for a quantitative measurement of chiral photoionization delays because the latter depend on both laboratory-frame angles of photoemission ($\theta$ and $\phi$) in any chiral-sensitive experiment. Extended Data Fig. 4 illustrates this fact by showing that the phase of the SB oscillation linearly depends on $\phi$ in the case of co-rotating XUV and IR fields. This is the RABBIT analogue of the angular streaking principle. Taking this property into account allows us to perform a quantitative analysis of angle-resolved chiral photoionization delays. Figure 4a,b shows the measured $\theta$-resolved RABBIT traces of SB 8 for the two enantiomers of MeOx in the case of a co-rotating circularly polarized IR field, the configuration that yields the largest PECD enhancement in line with theoretical calculations. The RABBIT fringes are not vertical but show a tilt as a function of $\theta$, in a direction that inverts when exchanging the enantiomers, again demonstrating the chiral nature of the effect. The time delays extracted at each $\theta$ angle (relative to $\theta = 90°$) are shown in Fig. 4c. They vary by about 240 as over the investigated angular range. Figure 4d,e shows the calculated angle-resolved RABBIT traces for the two enantiomers, respectively, closely reproducing the opposite tilts of the RABBIT fringes for the two enantiomers. The angle-resolved photoionization delays shown in

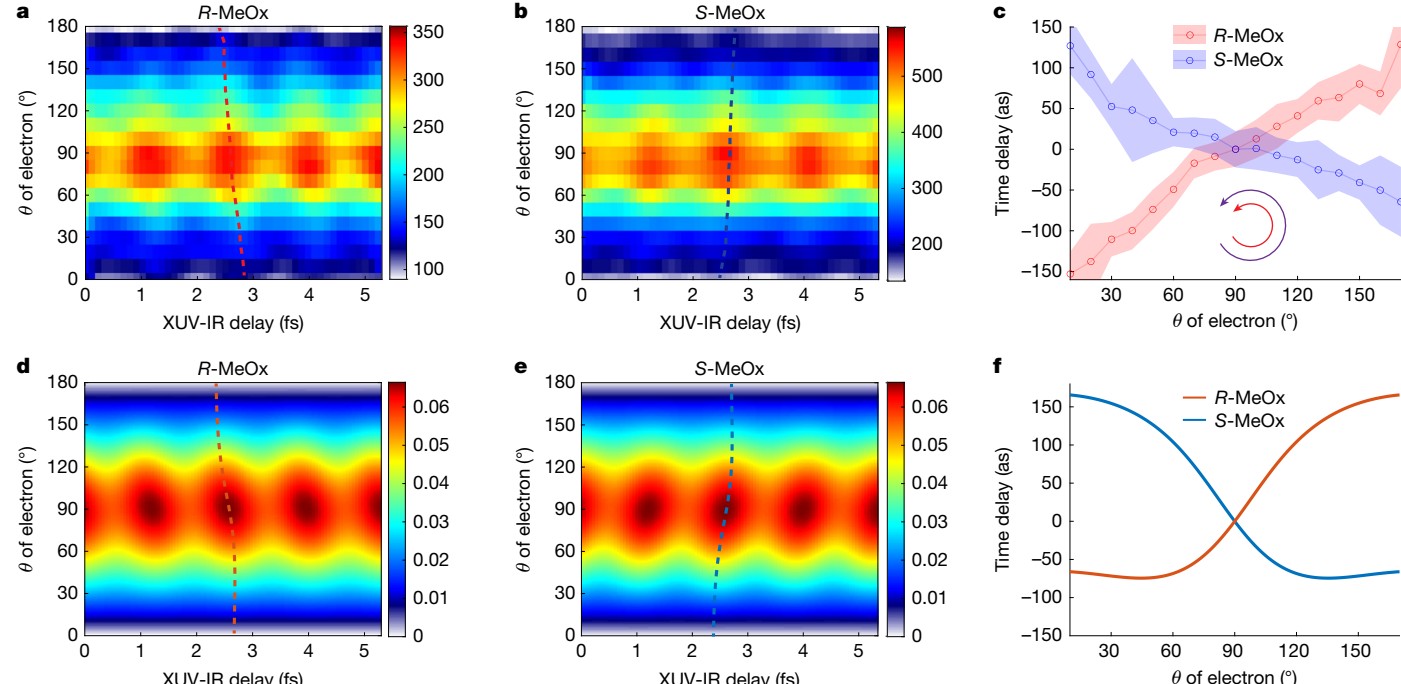

**Fig. 4 | Angle-resolved photoionization time delays in co-rotating XUV + IR fields. a,b,** $\theta$-resolved RABBIT traces of SB 8 for the $R$ and $S$ enantiomers of MeOx. **c,** The extracted photoionization time delay from **a** and **b,** for which the delay values at $\theta = 90°$ were chosen as reference. Note that the negative value of the ionization time delay indicates that the photoelectron wave packet is delayed with respect to emission at $\theta = 90°$, chosen as the reference in these measurements. The $\theta$-dependent maxima are highlighted in **a** and **b** as dashed lines and the equivalent lines are shown in **d** and **e.** The uncertainty of the extracted time delay (error bar) is estimated by the background-over-signal approach[34]. **d–f,** The corresponding results from theoretical calculations described in the text and in Methods.

Fig. 4f closely reproduce the measured variation of the photoionization delays by about 240 as over the measured angular range. The remaining differences between the theoretical results and the experimental data may originate from non-dipole effects, non-perturbative effects in the light–molecule interaction or electronic correlation effects. The experiment was performed well within the perturbative regime and the results shown in Fig. 2a,b do not provide evidence for strong non-dipole effects. We therefore expect electronic correlations to be the primary cause for deviations between experiment and simulations.

The comparison of angle-resolved time delays obtained with circularly or linearly polarized IR pulses also provides insights into the contribution of the continuum–continuum transitions to the chiral asymmetries of the time delays. Extended Data Fig. 5 shows such a direct comparison, with panels a and b displaying the raw experimental data for the case of a linearly polarized IR field and panels c and d comparing the angle-dependent photoionization time delays for linearly and circularly polarized IR fields, respectively. Whereas the delay variation amounts to approximately 180 as in the case of a linear IR field, it reaches approximately 240 as in the case of a circularly polarized IR field. To quantify the chiral asymmetry contributed by the continuum–continuum transitions, panel e shows the symmetrized time delays for a given polarization configuration (obtained as $(\tau(\theta)_{S\text{-MeOx}} + \tau(180° - \theta)_{R\text{-MeOx}})/2)$ and panel f shows their difference, which quantifies the chiral contribution of the continuum–continuum transitions to the measured photoionization delays. Although the error ranges overlap with zero, a clear trend can be recognized, corresponding to a variation of roughly 60 as over the whole angular range that we assign to the chirality of the IR-induced continuum–continuum transitions in the chiral potential.

A comparison of the present results, obtained with circularly polarized attosecond pulses in the perturbative regime, with previous results, obtained with femtosecond pulses in the strong-field regime[17,20,22], further highlights the achieved advances. The perturbative nature of the present approach results in a transparent control mechanism and good agreement with calculations. In the present work, the naturally occurring PECD effect has been enhanced by a factor of two, reaching up to 16%, and has been coherently controlled on the attosecond timescale in the perturbative regime involving pathways driven by one XUV and one IR photon each. Using femtosecond pulses, Rozen et al. observed asymmetries of up to 0.5% and their coherent control using non-perturbative two-colour strong-field ionization, but a detailed understanding of the mechanism as well as agreement with theory were missing[20]. In this work, we have measured chirality-sensitive photoionization delays by advancing the established RABBIT technique to circularly polarized attosecond pulses. The photoionization delays have been measured with 3D momentum resolution, in coincidence with specific ionic fragments and resolved in terms of the cationic final states. Good agreement with theory and a transparent explanation of the underlying mechanisms have been provided. By contrast, Beaulieu et al.[17] have used strong-field ionization based on non-perturbative two-colour femtosecond pulses. They have measured the projection of the PAD on a 2D detector parallel to the propagation direction of the laser pulses. This geometry does not resolve the emission angle in the polarization plane of the laser pulses on which the photoionization delay depends. Moreover, the photoionization delays determined in two-colour strong-field ionization depend on the intensity of the laser pulses[29], such that they do not represent intrinsic molecular properties.

In conclusion, we have introduced attosecond chiroptical spectroscopy and applied it to resolve and control the photoionization dynamics of chiral molecules on their natural, attosecond timescale. Our study demonstrates the powerful capability of the combination of circularly polarized attosecond light pulses with the 3D-momentum-resolved electron-ion coincidence detection, which allows examination of the attosecond electron dynamics induced by photoionization in an electronic-state-resolved, angle-resolved, energy-resolved and enantiomer-resolved manner. Electron scattering in a chiral molecular

potential gives rise to the asymmetric scattering amplitudes with respect to the light-propagation direction, underlying the PECD phenomenon, which has high chiral sensitivity and broad applicability. Attosecond chiroptical spectroscopy has now also revealed that one-photon-ionization delays of chiral molecules also show chiral asymmetries, defining a general approach to time resolve the electron-scattering dynamics in chiral molecules. Attosecond chiroptical spectroscopy also has the potential of answering a very important question about the origin of the chirality-induced spin selectivity (CISS) effect[30], which still lacks a quantitative explanation. Whereas traditional approaches view CISS as a purely electronic (spin–orbit) effect, such models underestimate the CISS effect by typically two orders of magnitude[31]. Very recent models suggest that coupled electronic-nuclear dynamics could play a central role[32]. Attosecond chiroptical spectroscopy may offer a solution to this intriguing puzzle through its ability of temporally separating electronic from structural dynamics[33].

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

# Methods

## Experimental methods

**Experimental set-up.** Near-IR laser pulses (2 mJ) were delivered from a regenerative Ti:sapphire laser amplifier at a central wavelength of 800 nm with the repetition rate of 5 kHz. The pulse duration of 25 fs (full width at half maximum (FWHM) in intensity) was measured by a home-built transient-grating FROG apparatus. This laser beam was split with a 70:30 beam splitter and the more intense part was sent through our beam-in-beam module[1] and then focused by a silver mirror ($f$ = 45 cm) into a 3-mm-long, xenon-filled gas cell to generate a circularly polarized XUV APT through high-harmonic generation. Our beam-in-beam module consists of a specially designed half waveplate and a common quarter waveplate. The half waveplate is composed of two half discs and the fast-axis angles of the two parts are oriented at ±22.5° with respect to the horizontal boundary, respectively. After passing through this half waveplate, the polarization directions of the upper part and the bottom part of the beam are orthogonal to each other. The two parts of the beam are then converted to be circularly polarized but with opposite helicities by transmission through the quarter waveplate[1]. The left-circularly polarized XUV beam in the two produced beams was picked up using a perforated mirror and focused into the main chamber of a COLTRIMS set-up by a nickel-coated toroidal mirror ($f$ = 50 cm). The XUV spectrum was characterized with a home-built XUV spectrometer consisting of an aberration-corrected flat-field grating (Shimadzu 120 lines per millimetre) and a micro-channel-plate detector coupled to a phosphor screen. The beam with 30% energy was used as the dressing field in RABBIT experiments. The dressing IR field was adjusted to linear polarization or left-circular polarization by a zero-order quarter waveplate and its intensity was controlled at a very low level (about $10^{11}$ W cm$^{-2}$) by an iris. The dressing IR beam was focused by a perforated lens ($f$ = 50 cm) and then recombined with the XUV beam by a perforated mirror. In the arm of the dressing field, there were two delay stages, that is, a high-precision direct-current motor (PI, resolution 100 nm) and a piezoelectric motor (PI, resolution 0.1 nm), adjusting the interpulse delays on femtosecond and attosecond timescales, respectively. A HeNe continuous-wave laser beam was sent through the beam splitter and traced the IR path in the two arms of the interferometer. A fast CCD camera behind the recombination mirror was used to lock the phase delay between the two arms through a PID feedback. When scanning the XUV-IR phase delay in the measurements, the piezo delay stage was actively stabilized at the step size of 110 as with a time jitter of less than 30 as.

**Sample delivery and detection.** The enantiopure MeOx samples were purchased from Sigma-Aldrich and contained in a heatable bubbler (about 0.5 bar at 30 °C). The supersonic gas jet was delivered along the $x$ direction through a small nozzle with an opening hole of 30 μm diameter and passed through two conical skimmers (Beam Dynamics) located 10 mm and 30 mm downstream with diameters of 0.2 mm and 1 mm, respectively. For the COLTRIMS spectrometer, static electric (about 2.21 V cm$^{-1}$) and magnetic (about 5.92 G) fields were applied along the $y$ axis to collect the charged fragments (electrons and ions) in coincidence. We did not observe any Coulomb explosion channels in our photoion–photoion coincidence (PIPICO) spectrum. Only the single ionization channels (one electron is coincident with one ion) were analysed and presented in this work.

**Data analysis.** In circularly polarized IR fields, the photoelectron azimuthal angle $\phi$ plays the same role as the XUV-IR delay[1] owing to the principle of angular streaking[35], as illustrated in Extended Data Fig. 4, and therefore the azimuthal angle should be fixed within a small range to extract the RABBIT phase or time-delay information. Otherwise, the time-delay information will be smeared out to the extent of averaging over $\phi$. To decrease the systematic errors in CD spectra as much as possible, we regularly switched the chiral sample from $R$ to $S$ enantiomer for each group of comparative experiments under the same laser conditions. Notably, the phase locking between XUV and IR beams was not interrupted when switching the sample, such that the measured chiral photoionization time delays for two enantiomers were referenced to the same time zero. To accurately extract the forward–backward photoionization time delay, we used Fourier transformation to isolate the $2\omega$ components in the yield oscillation of SBs. Extended Data Fig. 6a shows the raw data of the SB 8 yield as a function of the XUV-IR delay and panel b shows the Fourier intensity spectrum of panel a. An inverse Fourier transformation of the $2\omega$ oscillation signal is used to remove the DC background. The resulting data are shown in Fig. 3d, which allows us to intuitively compare the phases of the yield oscillations. For the delay-resolved PECD shown in Fig. 2d and Extended Data Fig. 2, we only keep the zero-frequency (DC) and $2\omega$ components and then perform the inverse Fourier transformation. We use the definition $2(I_S(\tau, \theta) - I_R(\tau, \theta))/(I_S(\tau, \theta) + I_R(\tau, \theta))$ to calculate the time-resolved PECD, in which $I_{S/R}(\tau, \theta)$ is the PAD after Fourier filtering at the XUV-IR delay of $\tau$ for the $S/R$ enantiomer. The energy integration window is 0.5 eV for each SB. For fitting the single-colour $\theta$-resolved PECD, we use the first-order spherical harmonic function $b_1^{1ph} \cos\theta$ to determine the asymmetry parameter $b_1^{1ph}$. For fitting the two-colour $\theta$-resolved PECD, we used the fitting function $b_1 \cos(\theta) + b_3 \cos^3(\theta)$ as derived in the 'Theoretical methods' section, which contains the higher-order term $b_3$. The PECD value at $\theta_k = 0$ is $b_1 + b_3$ in the two-colour case and $b_1^{1ph}$ in the one-colour case. Therefore, we use $b_1 + b_3$ to represent the two-colour PECD values shown in Fig. 2c,d and Extended Data Fig. 2. All angle-resolved and angle-integrated RABBIT traces demonstrated in this work show the absolute electron counts. To generate the PECD plots, we normalized each PAD by its own maximum, because our measurement conditions are unlikely to be stable enough to resolve the absolute CD on the total photoelectron counts, which is a non-dipole effect and beyond the scope of our manuscript.

## Theoretical methods

**Computational model.** For the simulation of the angle-resolved photoelectron spectra, we use the framework of refs. 3,4,27. Briefly, we describe the electronic states of the molecule in the frozen-core Hartree–Fock approximation[36–39] while treating the light–molecule interaction in the electric-dipole approximation. Hartree–Fock orbitals have been obtained with MOLPRO[40,41]. The field-free continuum electronic states have been calculated with the ePolyScat program package[37–39], using partial waves up to $L_{max}$ = 20 for the single-centre expansion of the molecular potential, which was found to yield sufficiently converged transition matrix elements for photoelectron energies up to 20 eV. By comparison, $L_{max}$ = 40 has been used for CHBrClF in ref. 4. We use second-order time-dependent perturbation theory to simulate the photoionization dynamics starting from the HOMO. For the field intensities used in the experiment and the simulations, this is sufficient to describe the interference between the relevant ionization pathways[3,4,27]. A particular challenge is the calculation of transition matrix elements between electronic continuum states. Because application of the dipole operator localizes the perturbed ionized states, we can converge the representation of these states with respect to the Gaussian basis set, using the unoccupied orbitals from the Hartree–Fock calculation as intermediate states[4]. However, this approach has the caveat that the discretization of the continuum may introduce artificial resonances that strongly disturb the photoelectron spectrum. We use two strategies to circumvent this problem. (1) To achieve a dense sampling of the photoelectron continuum, we choose the large Gaussian basis set aug-cc-pVQZ (ref. 42) and amend it by primitive diffuse $s$, $p$ and $d$ Gaussian functions with an exponent coefficient of 0.015 atomic units centred at the centre of charge of the molecule, as done in ref. 43. (2) To remove residual artificial resonances, we fit the virtual Hartree–Fock energies and optimize the slope of the fitted energies, to minimize

resonance effects. The artificial resonances not only affect the intensity of the SB signals but also strongly influence the PECD and RABBIT traces. By choosing the slope of the fitted virtual energies so as to avoid resonance-like enhancements of individual SBs, we achieve good agreement with the experimentally found forward–backward delays shown in Fig. 3. Establishing a more detailed understanding of the effects of the artificial resonances is the subject of continuing research. The XUV pulse is modelled as a frequency comb comprising the harmonics H7, H9, H11, H13 and H15 based on a fundamental photon energy of 1.55 eV without a temporal chirp. All XUV pulses have a Gaussian-shaped intensity profile with a FWHM of 5 fs. An IR pulse oscillating at the fundamental frequency with the same intensity profile as the individual XUV pulses but two times their intensity has been applied without delay to the XUV pulses. The FWHM of the IR pulse intensity profile is 5 fs. The parameters of the pulses are chosen to best reproduce the widths and relative intensities of the experimentally measured photoelectron spectrum shown in Fig. 1d. Because the IR pulse used in the experiment is much longer than in the simulation, we fix the delay between the XUV and IR pulses and instead vary the carrier envelope phase of the IR pulse.

**Angle and delay dependence of PECD.** The perturbative treatment of the photoionization results in the usual expressions for the orientation-averaged laboratory-frame PAD and PECD (see refs. 3,4,27 for details). At the peaks in the photoelectron spectrum generated by the harmonics of the XUV comb, the PAD has the form

$$PAD_H(k, \theta_k, \phi_k) = \sum_{L=0}^{2} \sum_{M=-L}^{L} \beta_{LM}^{1ph}(k) Y_{LM}(\theta_k, \phi_k).$$

The one-photon expansion parameters, $\beta_{LM}^{1ph}$, are independent of the XUV-IR delay. In the case of a circularly polarized XUV, only the terms with $M = 0$ are non-zero and the PECD takes, up to normalization depending only on even powers of $\cos\theta_k$, the well-known form

$$PECD_H(k, \theta_k) = \beta_{10}^{1ph}(k) \sqrt{\frac{3}{\pi}} \cos\theta_k = b_1^{1ph}\cos\theta_k,$$

in which $k$, $\theta_k$ and $\phi_k$ are the spherical polar coordinates of the photoelectron momentum.

At the SBs, the angular distribution for the pulse configurations used in the experiments reads

$$PAD_S(k, \theta_k, \phi_k) = \sum_{L=0}^{4} \sum_{M=0,\pm2} \beta_{LM}^{2ph}(k) Y_{LM}(\theta_k, \phi_k). \tag{1}$$

Under enantiomer exchange, the expansion parameters change sign according to $(-1)^L \beta_{LM}^{2ph}$. Hence, the expression for the PECD at the SBs is, up to normalization depending only on even-$L$ terms, given by

$$PECD_S(k, \theta_k, \phi_k) = \sum_{L=1,3} \sum_{M=0,\pm2} 2\beta_{LM}^{2ph}(k) Y_{LM}(\theta_k, \phi_k). \tag{2}$$

When $\phi_k = 0$, this expression can be reduced to

$$PECD_S(k, \theta_k) = b_1\cos\theta_k + b_3\cos^3\theta_k,$$

in which the asymmetry parameters $b_1$ and $b_3$ are simple functions of $\beta_{LM}^{2ph}$. The PECD value at $\theta_k = 0$ is $b_1 + b_3$ in the two-colour case and $b_1^{1ph}$ in the one-colour case. Therefore, we use $b_1 + b_3$ to represent the two-colour PECD values shown in Fig. 2c,d and Extended Data Fig. 1.

The two-photon anisotropy parameters at the SBs contain interfering contributions from the two ionization pathways through the neighbouring harmonics. For co-rotating or counter-rotating XUV and IR pulses, this pathway interference gives rise to the terms with $M = \pm 2$, which oscillate as function of the XUV-IR delay with frequency $2\omega_{IR}$, in which $\omega_{IR}$ is the frequency of the IR pulse. The $M = 0$ terms only contain

the single-pathway contributions and do not oscillate with the delay. By contrast, an IR pulse linearly polarized in the polarization plane of the XUV pulse gives rise to both interfering and non-interfering contributions for both $M = 0$ and $|M| = 2$. These, however, differ in their delay dependence: for $M = 0$, the interference terms oscillate with $2\omega_{IR}$, whereas the single-pathway contributions do not oscillate. For $|M| = 2$, the situation is reversed. In the angle-resolved PECD, $M = 0$ terms are maximal at $\theta_k = 0°$ and 180°, whereas terms with $|M| = 2$ are maximal at $\theta_k \approx 55°$ and 125°. Changing IR polarization and delay thus enables control over the anisotropy in the angle-resolved PECD. As both the PAD from equation (1) and the PECD from equation (2) contain interference terms, the normalized PECD can show higher-order modulations of the $2\omega_{IR}$ oscillations. We extracted RABBIT traces and time delays from the angle-resolved photoelectron spectra with the same procedure as applied to the experimental data.

## Data availability

The data generated and analysed in this study are available on the public ETH data repository at https://doi.org/10.3929/ethz-b-000741781.

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

**Acknowledgements** We thank A. Schneider and M. Seiler for their technical support. This work was supported by ETH Zürich and the Swiss National Science Foundation through projects 200021 172946 and the NCCR-MUST, by the Deutsche Forschungsgemeinschaft project no. 328961117—SFB 1319 ELCH (Extreme light for sensing and driving molecular chirality). The computing for this project was performed on the Beocat Research Cluster at Kansas State University, which is financed in part by NSF grants CNS-1006860, EPS-1006860 and EPS-0919443, ACI-1440548, CHE-1726332 and NIH P20GM113109 and used resources of the National Energy Research Scientific Computing Center (NERSC), a U.S. Department of Energy Office of Science User Facility operated under contract no. DE-AC02-05CH11231 using NERSC award BES-ERCAP0024357. M.H. acknowledges funding from the European Union's Horizon 2020 research and innovation programme under the Marie Sklodowska-Curie grant agreement No 80145- FP-RESOMUS for the experimental part of this work carried out at ETH Zürich. His further work on this project, including data analysis, was supported by the Chemical Sciences, Geosciences, and Biosciences Division, Office of Basic Energy Sciences, Office of Science, U.S. Department of Energy, under Grant No. DE-FG02-86ER13491. C.A. and L.G. were supported by the Grant No. DE-SC0022105 from the same funding agency. C.A. also acknowledges NSF grant no. 2244539 for further support during the summer of 2023. A.B. and C.P.K. acknowledge financial support from the Deutsche Forschungsgemeinschaft (CRC 1319).

**Author contributions** H.J.W. and M.H. conceived the study. M.H. and J.-B.J. performed the experiments and analysed the data. Simulations were implemented and carried out by A.B., R.E.G. and C.A., with the supervision of L.G. and C.P.K. H.J.W. supervised the experimental part of the project. M.H. and H.J.W. wrote the paper, with the input of all co-authors.

**Funding** Open access funding provided by Swiss Federal Institute of Technology Zurich.

**Competing interests** The authors declare no competing interests.

**Additional information**
**Correspondence and requests for materials** should be addressed to Meng Han or Hans Jakob Wörner.

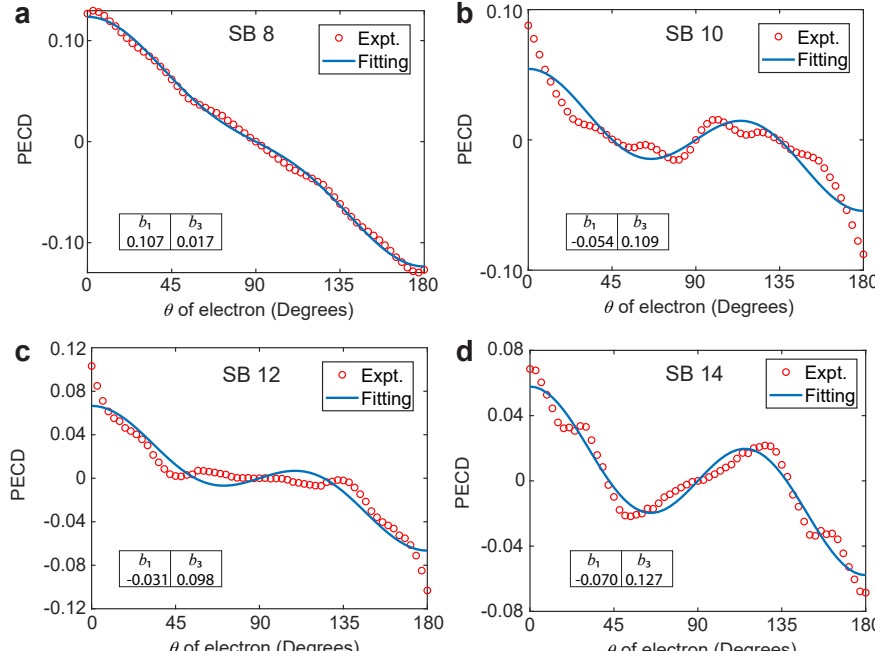

**Extended Data Fig. 1 | Determination of asymmetry parameters in two-colour photoionization. a–d**, Angle-resolved PECD distributions at SB positions, which are extracted from Fig. 2b. By fitting the angle-resolved PECD distributions with the two-colour formula shown in the 'Theoretical methods' section, we can extract the two-colour two-photon asymmetry parameters in the laboratory frame, which are shown in the left-lower corner of the panels.

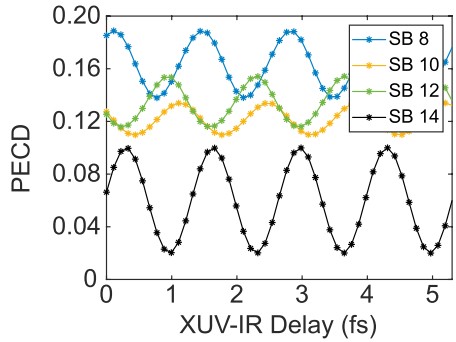

**Extended Data Fig. 2 | PECD obtained with circularly polarized IR field, co-rotating with the XUV field.** Delay-resolved PECD of SBs in the circularly polarized, co-rotating IR and XUV fields.

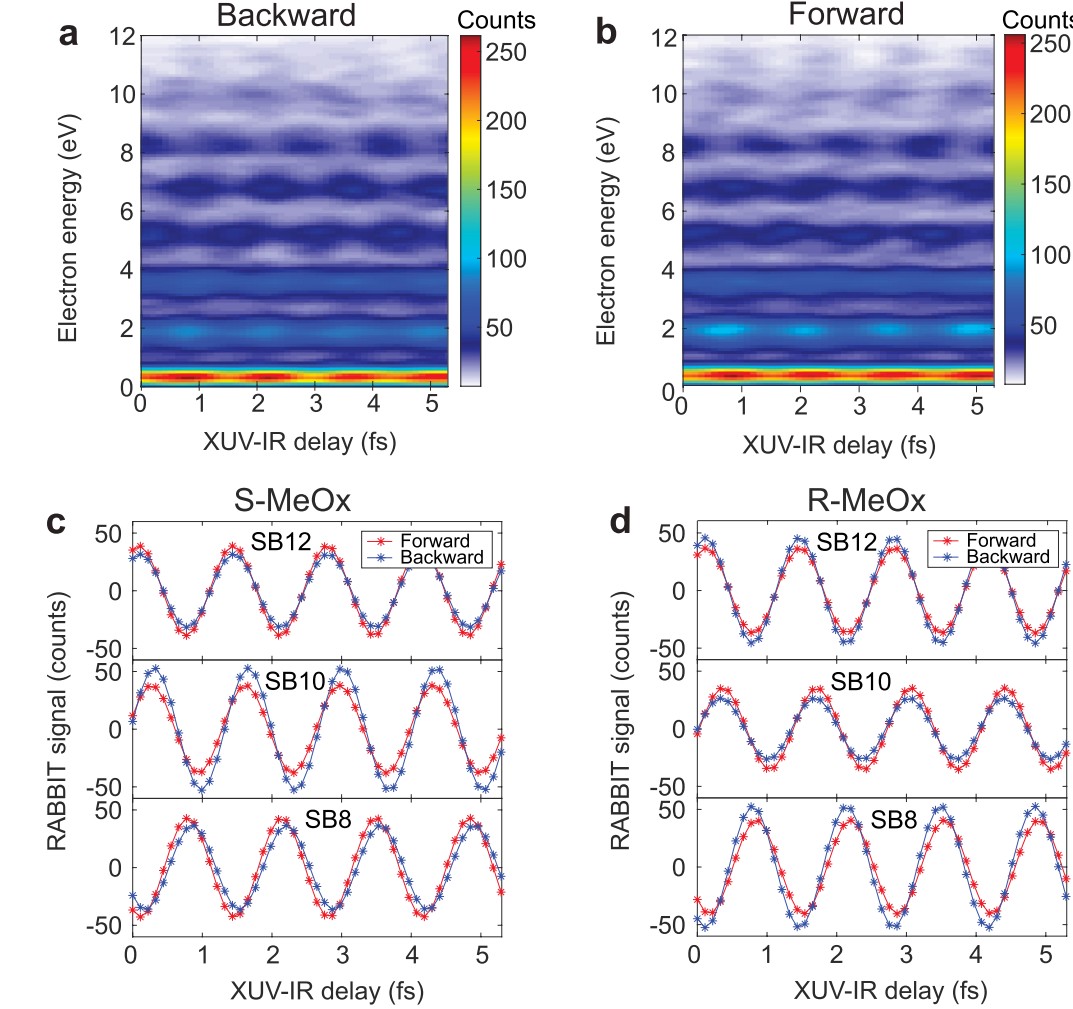

**Extended Data Fig. 3 | Experimental data for *S* enantiomer of MeOx in the linearly polarized IR field. a,b,** Angle-integrated RABBIT traces for backward and forward photoelectrons from the *S* enantiomer, respectively. **c,** The extracted yield oscillations of the SBs from **a** and **b. d,** The corresponding results from the *R* enantiomer, which are also shown in Fig. 3d.

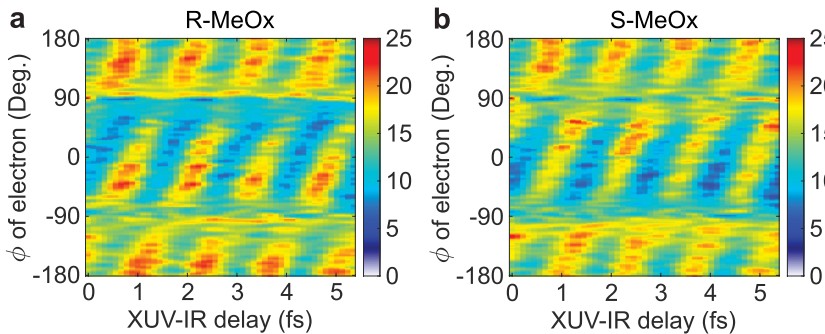

**Extended Data Fig. 4 | φ-resolved RABBIT traces. a,b,** φ-resolved RABBIT traces of SB 8 in the circularly polarized IR field for the *R* and *S* enantiomers of MeOx, respectively. Owing to the angular streaking effect, photoelectrons at different φ angles will have different RABBIT phases. Therefore, the φ angle of photoelectrons must be resolved to correctly measure the photoionization time delay. Note that there is no angular resolution at φ = ±90° owing to the static magnetic field used in the COLTRIMS spectrometer.

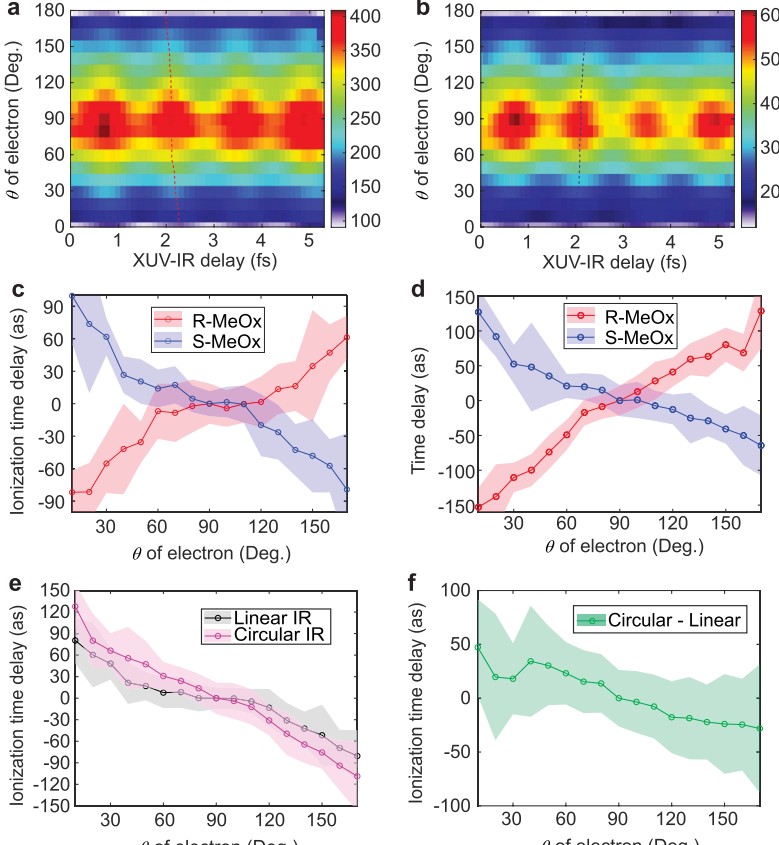

**Extended Data Fig. 5 | Two-colour angle-resolved photoionization time delay. a,b**, $\theta$-resolved RABBIT traces of SB 8 in the linearly polarized IR field for the $R$ and $S$ enantiomers of MeOx, respectively. **c**, The extracted photoionization time delay from **a** and **b**, for which the delay values at $\theta = 90°$ were chosen as the reference. **d**, The corresponding photoionization time delay in the circularly polarized IR field, which is already shown in Fig. 4c. **e**, The symmetrized time delay between two enantiomers, that is, $(\tau(\theta)_{S\text{-MeOx}} + \tau(180° - \theta)_{R\text{-MeOx}})/2$. **f**, Time-delay difference between circular and linear polarizations, isolating the chiral part of the cc delays.

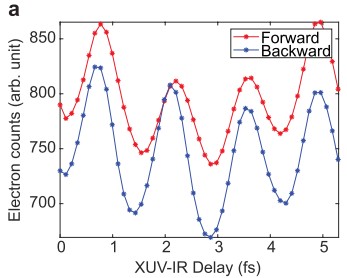

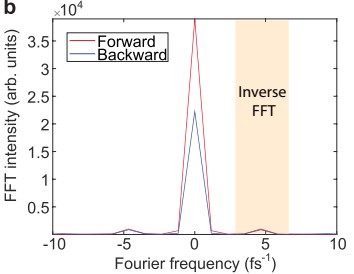

**Extended Data Fig. 6 | Fourier analysis of the experimental data. a**, Raw data of the SB 8 yield as a function of the XUV-IR delay. **b**, Fourier intensity spectrum of the data shown in **a**, for which the range of the $2\omega$ oscillation signal has been highlighted in orange. An inverse Fourier transformation of the $2\omega$ oscillation signal is used to remove the DC background. The resulting data are shown in Fig. 3d.