## [Peer Review file · Nature]

Attosecond Control and Measurement of Chiral Photoionisation Dynamics

Corresponding Author: Professor Hans Jakob Wörner

Version 0:

Reviewer comments:

Referee #1

(Remarks to the Author)

The manuscript entitled “Attosecond coherent control and measurement of the photoionization dynamics of chiral molecules” written by Meng Han et al. reports on RABBITT and PECD measurements with a circularly polarized attosecond pulse train and a linearly or circularly polarized IR dressing field in the two enantiomers of the chiral molecule methyloxirane (MeOx). The measurements are performed in a COLTRIMS apparatus that allows measuring the photoelectrons in coincidence with the parent ions. The authors observe that the PECD value, that is the forward/backward asymmetry in the emission of photoelectrons with respect to the light propagation axis, is higher with two color XUV + IR ionization than with single photon XUV ionization. The PECD is found to depend on the XUV-IR delay and to oscillate on the sub-cycle timescale. Then the oscillations of the sidebands signal in the forward and backward directions is analyzed to extract a photoionization time delay, of opposite value for the two enantiomers and decreasing towards zero as the photoelectron energy increases. Finally with co-rotating XUV + IR fields this photoionization time delay is found to strongly depend on the photoelectron emission angle. A difference is measured between the time delays with linear or circular IR, which is attributed to a sensitivity of the continuum-continuum transitions to the chiral potential. The measurements are compared to simulations but their assessment is out of the scope of my expertise.

The novelty of the work resides in the use of circularly polarized XUV attosecond pulse trains to perform RABBITT experiments in a chiral molecule. Nevertheless, the observed chiral effects and some of the conclusions have already been demonstrated with circularly polarized femtosecond visible or UV pulses (Beaulieu et al. *Science* 358, 1288-1294 (2017) – Ref. 18), structured electric fields (Rozen et al. *Phys. Rev. X* 9, 031004 (2019) doi: 10.1103/PhysRevX.9.031004) or synchrotron radiation (Garcia et al. *Phys. Chem. Chem. Phys.* 16, 16214-16224 (2014) – Ref 25 and e.g. Lehman et al. *J. Chem. Phys.* 139, 234307 (2013) doi: 10.1063/1.4844295), as will be detailed below. The attosecond pulse train structure does not bring fundamentally new insight compared to Ref. 18, even though the experiment performed in the present manuscript is elegant and challenging. Furthermore, the manuscript has multiple flaws, which I will describe below. Therefore I do not recommend its publication in Nature.

The authors write on p.1 that their work “demonstrate[s] attosecond coherent control over PECD”. Rozen et al. *Phys. Rev. X* 9, 031004 (2019) measured the forward/backward asymmetry in the photoelectron distribution of chiral molecules ionized by synthetic light fields, and showed that the PECD can be coherently controlled with the relative delay of the two colors on the sub-cycle (i.e. attosecond) time scale. See e.g. Figure 3 (a) to (f) of that reference where the delay between the two colors is varied with 333 as steps.

The authors claim that their work “demonstrate[s] the measurement of chiral asymmetries in the photoionization delays of chiral molecules”. Such chiral asymmetries have been measured already by Beaulieu et al. (Ref. 18) in camphor, as highlighted by the comparison of Figure 3C of Ref. 18 with Figure 3f of the present manuscript. A similar trend is observed in both cases, the forward/backward time delay decreases with increasing electron kinetic energy. As noted by the authors of the present manuscript, the time delays measured here for SB8 exceed the time delays measured in Ref. 18, but let us note that in Ref. 18 the lowest energy sideband was resonant with an autoionizing state and showed a very different behavior compared to the other sidebands. If one compares the time delays for the same electron energy in the two experiments, they are of similar magnitude considering the error bars.

The similarities between the results reported here and those of Ref. 18 are also visible if one considers the angle-resolved

photoionization time delays, comparing Figure 3D of Ref.18 and Extended Data Figure 4c of the present manuscript. The effect appears stronger in MeOx than in camphor but is qualitatively similar. Here again let us note that the MeOx results are for an electron kinetic energy of ~2 eV while the camphor results of Ref. 18 concern ~5 eV electrons which are probably less sensitive to the chiral potential, a difference that could explain partly the difference in magnitude of the time delays. Finally, Ref. 18 also changed the polarization of the IR dressing field from linear to circular to separate the chiral contributions of the electronic and continuum-continuum transitions. In the case of circularly polarized IR, the angularly-resolved measurement shows a chiral asymmetry attributed to the influence of the chiral potential on the continuum-continuum transitions. This is the effect discussed by the authors here in Extended Data Figure 4 and on page 11. Another result highlighted by M. Han et al. in this manuscript is the double PECD value in the case of XUV + IR ionization compared to XUV-only ionization. Since its experimental demonstration, PECD has been studied with various ionization schemes: single photon, multiphoton, REMPI, tunnel, etc. As PECD arises from interferences between partial waves constituting the electron wave packet, it is well known that it is sensitive to the intermediate state(s) involved. The magnitude of PECD in a same system thus depends on the ionization scheme. For example, Lehman et al. *J. Chem. Phys.* 139, 234307 (2013) demonstrated that the PECD in camphor using femtosecond multiphoton ionization at 400 nm was 60% higher in magnitude and of opposite sign compared to the value measured with single photon ionization at a synchrotron. Therefore I do not think that the result presented here in Figure 2c is novel enough to deserve publication in *Nature*. In addition, the authors do not show in Fig. 2c the PECD values at the harmonics' energies in the XUV+IR case, which would be a more direct comparison with the XUV-only case. In Ref. 4 the PECD vanishes at the harmonics peaks. Finally, the authors discuss in Figure 1d and 1e the photoelectron spectra measured in coincidence with MeOx ion or fragments. Their analysis shows that the photoelectron spectrum of the parent ion is due to ionization of the HOMO orbital only (X ionic state), while the A and B ionic states contribute to the C₂H₄⁺ (m/q=28) channel. These aspects of MeOx photoionization spectroscopy are actually known from the synchrotron measurements of Garcia et al. *Phys. Chem. Chem. Phys.* 16, 16214-16224 (2014) (Ref.25), see Figure 4 of that reference.

I also have concerns with a few technical aspects of the paper. First, the article relies heavily on Reference 4 (cited on pages 1, 5, 7, 9 and 11), which is not a peer-reviewed article.

Second and most importantly, the authors define the PECD as $2[IS(E_k, \theta) - IR(E_k, \theta)]/[IS(E_k, \theta) + IR(E_k, \theta)]$ (page 5). Apart from the fact that this is not what the arrows on top of Figure 2a indicate (they suggest that the XUV polarization is rotated), this requires changing the enantiomer in the COLTRIMS "regularly" (how often?) while keeping the exact same experimental conditions. How do the authors take into account possible artifacts coming from differences between the two enantiomers purity/enantiomeric excess? between different pressures of the two molecules in the COLTRIMS? Are the data shown in Figure 2a-b symmetrized and/or antisymmetrized? Comparing the colorbars in the RABBITT traces for the two enantiomers in Figure 3a-c (R MeOx) and Extended Data Figure 3 a-b (S MeOx), one clearly sees that the S enantiomer resulted in higher counts. The authors state that they have verified that the chiral asymmetry is also present when switching the XUV helicity in the same enantiomer, but no data is presented to support this claim. A measurement in a racemic mixture of MeOx, or with a linearly polarized XUV attosecond pulse train, could also have been shown to strengthen the validity of the conclusions.

Third, the manuscript shows lacks of rigor in multiple points. It is unclear what is the harmonic spectrum used in the experiments. In Figure 1a, the spectrum shows harmonics H7 to H13, which is also written on page 3. But from Figure 2 on, the authors discuss SB14, suggesting interferences between H13 and H15. What is the actual spectrum used in the experiments? It also looks like the authors do not use the same definition of the forward-backward time delay throughout the manuscript (forward minus backward or backward minus forward), making the presented results inconsistent with each other. Indeed, in Figure 3d SB8, the forward oscillation is ahead of the backward oscillation and the time delay in Figure 3f is positive. In Extended Data Figure 4 a and c (which corresponds to the same experiment), the forward electrons correspond to theta between 0-90°, and the backward electrons to theta between 90-180°. The forward electrons oscillations are here again ahead of the backwards electrons (in agreement with Figure 3d), but in Extended Data Figure 4c, for R MeOx, if one calculates the time delay forward minus backwards it is now negative, in disagreement with Figure 3f.

Version 1:

Reviewer comments:

Referee #1

(Remarks to the Author)

The authors have added technical details on p.3 and p.11-12 of the manuscript highlighting the differences between their work and existing literature. As I wrote in my first review, they reside in the use of a COLTRIMS apparatus for the detection of the electrons in coincidence with ions and of an XUV attosecond pulse train for ionization. This is a different geometry of the detector and a different ionization regime compared to the literature. While this represents a challenging and elegant experiment, of interest to specialists, I do not think that it represents a conceptual advance for ultrafast photoionization of chiral molecules of interest to the broad readership of *Nature*. The fact that there is a delay between the electrons wavepackets emitted forwards or backwards in a chiral molecule, that is opposite for enantiomers, has been measured and calculated in Ref 18. The fact that a chiral signal in the photoionization of a chiral molecule can be controlled with light on the subcycle timescale has been shown experimentally and theoretically in Ref 29. The fact that the PECD values can differ in sign and magnitude (be "enhanced") depending on the ionization scheme (single photon XUV, multiphoton IR, multiphoton visible... two-photon XUV + IR in the present manuscript) has been thoroughly shown in the literature (e.g. *J. Chem. Phys.* 139, 234307 (2013) in Camphor).

The assessment of the calculations that are compared to the experimental results is out of my field of expertise. As in the previous submission, they still rely heavily on Ref 4 which is under review at another journal according to the authors. The authors have clarified technical points that I raised, but they were not the main reason why I did not recommend publication of this manuscript in Nature. The main reason being the lack of fundamental new insight regarding ultrafast photoionization of chiral molecules as detailed above, which still holds in the revised version. Therefore I do not recommend publication of this manuscript in Nature.

Referee #2

(Remarks to the Author)

The work of Meng Han et al. presented in "Attosecond Coherent Control and Measurement of the Photoionization Dynamics of Chiral Molecules" constitutes a leap forward in our capabilities to study attosecond chiral molecular dynamics. It must be said that this is not the first work on the subject, and as rightfully pointed out by another referee, one must properly acknowledge these pioneering works by Beaulieu et al. presented in Science (2017) and by Rozen et al. in Physical Review X (2019). Over the past years and decades, different strategies have been implemented to study how chiral effects depend on both structure and dynamics in molecules. The present work is elegant and it is the first "RABBIT" experiment with circularly polarized attosecond pulses aimed to resolve chiral properties of molecules in space and time. It shows that it is possible to enhance and coherently control chiral properties. Such control has been shown previously using other techniques. This means that we must ask ourselves if the adaptation of the "RABBIT" technique to this type of problem warrants publication in NATURE. I believe that it does because the RABBIT scheme will help to transform this research field from a phenomenological field of laser-induced dynamics to a quantitative and systematic research field for studies of chirality of molecules in space and time.

Prior works attaining attosecond delays experimentally in chiral molecules are criticized by the authors, as lacking a proper theoretical basis for interpreting the experiments. There is a fine line here, and while I see the great value in these prior works, I believe that the authors are correct in their assessment that the RABBIT technique will open up for quantitative studies of molecular chiral dynamics that go beyond qualitative laser-driven observations. The strength of RABBIT, which has previously been used for attosecond pulse characterization and for understanding photoionization delays and correlation effects in atoms, is that it provides a platform to perform quantitative experiments that produce numbers that can be compared with numbers from theoretical models with different approximations. In this way, we can test models and gain a better understanding of the system under study.

The experimental results are interpreted using theoretical molecular simulations including continuum-continuum dynamics. The agreement between experiment and theory is convincing with similar trends observed in time delays over energy and electron-emission angle. Thus, the frozen core approximation seems adequate for the study. In summary, the usage of the RABBIT technique is the major selling point of the work, as it implies measurements of "intrinsic properties" of the chiral molecules in space and time.

Referee #3

(Remarks to the Author)

In the manuscript entitled "Attosecond coherent control and measurement of the photoionization dynamics of chiral molecules", the authors of Meng Han et al. reports measurements of attosecond time-resolved photoelectron spectrograms of the two enantiomers of the chiral molecule methyloxirane. Technically, as far as I know, this is the first time that circularly polarized attosecond extreme ultraviolet pulses have been used to explore ultrafast photoelectron dynamics. In addition, the three-dimensional momentum distribution of photoelectrons has been measured using COLTRIMS technology. Their experimental results not only achieved significantly enhanced photoelectronic circular dichroism (PECD) effect, but also enabled them to distinguish and understand PECD from more perspectives, such as electronic states, angles, energies, and enantiomers. Due to two-color two-photon ionization located in the perturbation regime, the time delay of photoemission measured by RABBIT type has a well-known and clear physical concept, namely the Wigner delay, which is associated with the inherent properties of the target molecule. However, two-color strong-field ionization locates in non-perturbative regime, involving multiphoton ionization and even tunneling ionization, with more complex photoionization dynamics and thus it is difficult to quantitatively explain the underlying mechanisms. There is no physical concept corresponding to the photoemission time delay measured based on this technology that can be understood. Therefore, the current work provides more complete theoretical supports for the attosecond measurement and control of chiral molecules. The current work enables the direct manipulation of photoelectron circular dichroism with attosecond - level precision, representing a significant advancement compared to previously published works (in contrast to femtosecond - scale precision). I believe that the current work represents a significant advancement in attosecond metrology using circularly polarized pulses, enabling attosecond - scale coherent control and measurement of the photoionization dynamics of chiral molecules. However, compared to Reference 4, aside from the experiments, there are no new conceptual discoveries or breakthroughs. If the current work can overcome this lack of conceptual innovation, it should be able to attract the interest of a wide range of researchers in the fields of attosecond science and chiral molecule research. In that case, I would be happy to recommend its publication in Nature.

Version 2:

Reviewer comments:

Referee #3

(Remarks to the Author)

The authors have elaborated on and revised the conceptual innovations of their work in great detail from multiple perspectives, satisfactorily addressing my questions. I hereby recommend that their work be published in Nature.

We thank the reviewer for their review and thoughtful comments. Below, we reproduce their statements in black, provide our replies in blue and the changes made to the manuscript in green.

Referees' comments:

Referee #1 (Remarks to the Author):

The manuscript entitled “Attosecond coherent control and measurement of the photoionization dynamics of chiral molecules” written by Meng Han et al. reports on RABBITT and PECD measurements with a circularly polarized attosecond pulse train and a linearly or circularly polarized IR dressing field in the two enantiomers of the chiral molecule methyloxirane (MeOx). The measurements are performed in a COLTRIMS apparatus that allows measuring the photoelectrons in coincidence with the parent ions. The authors observe that the PECD value, that is the forward/backward asymmetry in the emission of photoelectrons with respect to the light propagation axis, is higher with two color XUV + IR ionization than with single photon XUV ionization. The PECD is found to depend on the XUV-IR delay and to oscillate on the sub-cycle timescale. Then the oscillations of the sidebands signal in the forward and backward directions is analyzed to extract a photoionization time delay, of opposite value for the two enantiomers and decreasing towards zero as the photoelectron energy increases. Finally with co-rotating XUV + IR fields this photoionization time delay is found to strongly depend on the photoelectron emission angle. A difference is measured between the time delays with linear or circular IR, which is attributed to a sensitivity of the continuum-continuum transitions to the chiral potential. The measurements are compared to simulations but their assessment is out of the scope of my expertise.

The novelty of the work resides in the use of circularly polarized XUV attosecond pulse trains to perform RABBITT experiments in a chiral molecule. Nevertheless, the observed chiral effects and some of the conclusions have already been demonstrated with circularly polarized femtosecond visible or UV pulses (Beaulieu et al. *Science* 358, 1288-1294 (2017) – Ref. 18), structured electric fields (Rozen et al. *Phys. Rev. X* 9, 031004 (2019) doi: 10.1103/PhysRevX.9.031004) or synchrotron radiation (Garcia et al. *Phys. Chem. Chem. Phys.* 16, 16214-16224 (2014) – Ref 25 and e.g. Lehman et al. *J. Chem. Phys.* 139, 234307 (2013) doi: 10.1063/1.4844295), as will be detailed below. The attosecond pulse train structure does not bring fundamentally new insight compared to Ref. 18, even though the experiment performed in the present manuscript is elegant and challenging. Furthermore, the manuscript has multiple flaws, which I will describe below. Therefore I do not recommend its publication in *Nature*.

We thank the reviewer for describing our work as innovative and elegant. This is indeed the first experiment (to our knowledge) that studied chiral molecules with circularly polarized attosecond pulses. It thereby opens many exciting perspectives, two of which are realized in the present manuscript: coherent control of PECD and chirality-sensitive attosecond chronoscopy.

We respectfully disagree with the reviewer regarding the novelty of the insight compared to Ref. 18. That work measured the phase shifts of sidebands in two-color strong-field ionization, which are known to depend on the intensity of the ionizing pulse, such that they cannot reflect an intrinsic molecular property, such as a Wigner delay in photoionization (see e.g. Ref. 30). This was later confirmed through detailed theoretical analysis (PRA 104, 043113 (2021) and PRA 106, 053101 (2022)). In contrast, the present work measures

photoionization delays through an analogue of the RABBIT technique, which has a well-defined and well-established relationship to the intrinsic molecular property known as Wigner delays in photoionization. This key difference manifests itself in a detailed understanding of the XUV+IR photoionization dynamics, as shown by the good agreement between theory and experiment (Figs. 3 and 4 in our manuscript).

We have clarified this aspect in a new paragraph on p. 11-12.

The authors write on p.1 that their work “demonstrate[s] attosecond coherent control over PECD”. Rozen et al. Phys. Rev. X 9, 031004 (2019) measured the forward/backward asymmetry in the photoelectron distribution of chiral molecules ionized by synthetic light fields, and showed that the PECD can be coherently controlled with the relative delay of the two colors on the sub-cycle (i.e. attosecond) time scale. See e.g. Figure 3 (a) to (f) of that reference where the delay between the two colors is varied with 333 as steps.

It should first be pointed out that the forward-backward asymmetry studied by Rozen et al. is not PECD. PECD describes the fact that the differential cross section of the photoelectrons emitted in the forward hemisphere (along the propagation direction of the photoionizing light) is smaller or larger than into the backward hemisphere. The asymmetries studied by Rozen et al. vanish when integrated over each hemisphere. This is a consequence of the fact that they were not using circularly, but linearly polarized light. Therefore, Rozen et al. have studied a different type of asymmetry compared to PECD.

Second, Rozen et al. have used strong-field ionization to study forward-backward asymmetries in the photoelectron angular distribution as opposed to the perturbative regime used in the present work. The mechanism underlying the control of the forward-backward asymmetries in the work of Rozen et al. is very different compared to the perturbative regime of our work. In the work of Rozen et al., many different multi-photon pathways lead to the same final photon energies and contribute to the control mechanism. In contrast, in the perturbative regime of our work only two pathways contribute to each final state in the photoelectron spectrum, which offers a simple and transparent interpretation. The latter provides a much simpler relationship between the experimental observable (i.e., RABBITT phase) and the phase of the transition matrix element. This advantage motivated the development of circularly polarized attosecond pulses and the associated attosecond metrology.

Finally, it is worth noting that the observed asymmetries were much smaller in the work of Rozen et al.: up to 5×10^{-3} compared to 1.2×10^{-1} in our case. The realization of attosecond metrology with circularly polarized pulses is therefore not only a technical advance, but it also enabled the conceptual advance of realizing the coherent control and measurement of the photoionization dynamics of chiral molecules in the perturbative regime amenable to transparent physical interpretations.

These differences have been summarized in the new paragraph added on p. 11-12.

The authors claim that their work “demonstrate[s] the measurement of chiral asymmetries in the photoionization delays of chiral molecules”. Such chiral asymmetries have been measured already by Beaulieu et al. (Ref. 18) in camphor, as highlighted by the comparison of Figure 3C of Ref. 18 with Figure 3f of the present manuscript. A similar trend is observed in both cases, the forward/backward time delay decreases with increasing electron kinetic energy. As noted by the authors of the present manuscript, the time delays measured here for SB8 exceed the time delays measured in Ref. 18, but let us note that in Ref. 18 the lowest energy sideband

was resonant with an autoionizing state and showed a very different behavior compared to the other sidebands. If one compares the time delays for the same electron energy in the two experiments, they are of similar magnitude considering the error bars.

First, let us recall that “delays” (more precisely “phase shifts”) extracted from multiphoton ionization are intensity dependent as demonstrated in 2014 by Zipp et al. (Reference 30) who first applied the scheme later used by Beaulieu et al. to argon atoms and extracted intensity dependent delays. Second, the phase shifts measured in the presence of circularly polarized fields linearly depend on the azimuthal angle of ejection (“phi” in our manuscript) (see Reference 1). This is the angle of electron ejection in the polarization plane of the circularly polarized light. Resolving this angle is therefore crucial to access the “photoionization delays”. This angle is resolved in our measurements that detect the three-dimensional momentum distribution of the photoelectrons. The azimuthal angle is not resolved in the work of Beaulieu et al., because they used a velocity-map-imaging detector with a plane perpendicular to the polarization plane of the circularly polarized light. As a consequence of the broken symmetry, Beaulieu et al. therefore did not have access to data resolved as a function of the azimuthal angle. Therefore the “phase shifts” that they extracted are averaged over a large range of azimuthal angles, such that this observable contains both amplitude and phase contributions of the multiphoton transition pathways, which are not easily separated into pure phase contributions that would contain the desired information on the photoionization dynamics. This precludes the type of comparison suggested by the reviewer from the beginning.

We have removed the comparison of the magnitudes of time delays between the two works. We have added a comment regarding the importance of resolving the azimuthal angle of photoemission on p. 3.

Another crucial aspect in contrasting the present work with Ref. 18 is the use of electron-ion-coincidence detection. As illustrated in Fig. R1, coincidence measurements are essential for an accurate determination of attosecond photoionization delays. Figure R1 shows a comparison of photoelectron spectra with and without coincidence with ions in the XUV-only case. Without coincidence, it is impossible to determine the orbital from which photoionization occurs, due to the limited energy resolution. In Ref. 18, the sample was camphor (C₁₀H₁₆O), which has an even more congested photoelectron spectrum compared to our sample (although Ref. 18 does not show the single color, i.e. 400-nm only, photoelectron spectrum). Therefore, the results of Ref. 18 cannot be assigned to a single final state of the molecular cation, which again implies that the phase shifts measured in that work cannot be interpreted as photoionization delays.

Fig. R1. The upper four panels show the comparison of photoelectron spectra with and without coincidence with ions in the XUV-only case. The bottom panel is Fig.2D from Reference 18. This comparison illustrates the importance of electron-ion-coincidence spectroscopy to resolve individual final electronic states populated in the photoionization process.

The similarities between the results reported here and those of Ref. 18 are also visible if one considers the angle-resolved photoionization time delays, comparing Figure 3D of Ref.18 and Extended Data Figure 4c of the present manuscript. The effect appears stronger in MeOx than in camphor but is qualitatively similar. Here again let us note that the MeOx results are for an electron kinetic energy of ~ 2 eV while the camphor results of Ref. 18 concern ~ 5 eV electrons which are probably less sensitive to the chiral potential, a difference that could explain partly the difference in magnitude of the time delays.

The angle α , as a function of which the delays are resolved in Ref. 18 (Fig. 3D), is not identical with the polar angle θ in our work. The reason is, as explained in response to the previous point, the lack of azimuthal-angle (ϕ) resolution in Ref. 18. The signal measured under a given angle α in Ref. 18 is a convolution over a range of polar (θ) and azimuthal (ϕ) angles as defined in our work. This fact alone leads to a reduction of the phase shifts (or delays) as measured in Ref. 18. As a consequence, these phase shifts measured in Ref. 18 do not have a simple relation to the true phase shifts of photoemission in the laboratory frame (which depend on both θ and ϕ). In the present work, since COLTRIMS was used to measure the three-dimensional momenta of the electrons, the phase-shift (time-delay) information is resolved as a function of both θ and ϕ , which is necessary to convert the measured phase shift information to time delays.

We have removed the comparison of the numbers and have commented on the differences in angular resolutions of Ref. 18 and the present work.

Finally, Ref. 18 also changed the polarization of the IR dressing field from linear to circular to separate the chiral contributions of the electronic and continuum-continuum transitions. In the case of circularly polarized IR, the angularly-resolved measurement shows a chiral asymmetry attributed to the influence of the chiral potential on the continuum-continuum transitions. This is the effect discussed by the authors here in Extended Data Figure 4 and on page 11.

Although these aspects of the two works are indeed related, it should be emphasized that a rigorous separation of the chiral contributions of the photoionization and continuum-continuum contributions is only possible in the perturbative regime of RABBIT-type measurements and in the presence of three-dimensional momentum resolution. First, in the non-perturbative regime of strong-field ionization, it is not possible to distinguish between, e.g., a 4-photon-absorption pathway and a 5-photon-absorption-1-photon-emission pathway (referring to 2ω -photons only), such that there are no pure bound-continuum pathways. All pathways automatically contain continuum-continuum contributions. This is not the case in the perturbative RABBIT regime, where pure bound-continuum transitions (i.e. involving single XUV photons) become accessible. Second, the rigorous separation of chiral effects requires a rigorous separation of the polar and azimuthal angles of emission. This requires 3D momentum resolution, not achieved in Ref. 18, but realized in the present work.

Another result highlighted by M. Han et al. in this manuscript is the double PECD value in the case of XUV + IR ionization compared to XUV-only ionization. Since its experimental demonstration, PECD has been studied with various ionization schemes: single photon, multiphoton, REMPI, tunnel, etc. As PECD arises from interferences between partial waves constituting the electron wave packet, it is well known that it is sensitive to the intermediate state(s) involved. The magnitude of PECD in a same system thus depends on the ionization scheme. For example, Lehman et al. J. Chem. Phys. 139, 234307 (2013) demonstrated that the PECD in camphor using femtosecond multiphoton ionization at 400 nm was 60% higher in

magnitude and of opposite sign compared to the value measured with single photon ionization at a synchrotron. Therefore I do not think that the result presented here in Figure 2c is novel enough to deserve publication in Nature.

This argument of the reviewer basically expresses the opinion that the availability of single and multiphoton ionization schemes makes coherent control obsolete. We naturally disagree with this view. Whereas insights gained from the comparison of single- and multiphoton ionization to the same final energy are of course very valuable and interesting, the demonstration of coherent control over PECD is a major step forward. In addition to being a technical advance, it also creates a conceptual advance because the perturbative regime of our experiments enables a clear identification of the number of photons involved in each pathway, which in turn allows for a transparent theoretical description of the control mechanism. Our work demonstrates that PECD can be enhanced by a factor of ~ 2 over the entire range of studied photon energies and that it can reverse the sign of PECD by simply changing the XUV-IR delay on the attosecond time scale. This opens new perspectives for tailoring light fields to higher chiroptical discrimination and deepening our understanding of chiral electron scattering underlying PECD.

In addition, the authors do not show in Fig. 2c the PECD values at the harmonics' energies in the XUV+IR case, which would be a more direct comparison with the XUV-only case. In Ref. 4 the PECD vanishes at the harmonics peaks.

In Ref. 4, the PECD vanishes at the harmonic peaks because of the chosen normalization: the PECD data is shown relative to the XUV-only PECD, explaining the reviewer's observation. In the XUV+IR case, it is true that the PECD at the harmonic peaks will decrease or even vanish compared to the XUV-only case. This is because of the quantum interference of pathways, as discussed in detail in Ref. 4. However, the experimental data shown in Fig. 2c already illustrates the discussed enhancement effect on its own. Whereas the addition of a linearly polarized IR field does not (significantly) increase the PECD at the sideband positions compared to the averaged of the two neighboring mainband positions, the addition of a circularly polarized IR field does cause a significant increase.

The PECD value at $\theta_k = 0$ is $a_1 + a_3$ in the two-color case and $a^{\{\mathrm{ph}\}}_1$ in the one-color case. Note that we updated the two-color PECD values shown in Figs. 2C-D of the main text and Supplementary Material Figure 2. This update doesn't change our conclusions.

Finally, the authors discuss in Figure 1d and 1e the photoelectron spectra measured in coincidence with MeOx ion or fragments. Their analysis shows that the photoelectron spectrum of the parent ion is due to ionization of the HOMO orbital only (X ionic state), while the A and B ionic states contribute to the C₂H₄⁺ (m/q=28) channel. These aspects of MeOx photoionization spectroscopy are actually known from the synchrotron measurements of Garcia et al. Phys. Chem. Chem. Phys. 16, 16214-16224 (2014) (Ref.25), see Figure 4 of that reference.

We agree with the reviewer. Garcia et al (now Ref. 26) is indeed being cited in this context, both on the bottom of p. 3 and in the caption of Fig. 1.

I also have concerns with a few technical aspects of the paper. First, the article relies heavily on Reference 4 (cited on pages 1, 5, 7, 9 and 11), which is not a peer-reviewed article.

Reference 4 is currently under review in PRL.

Second and most importantly, the authors define the PECD as $2[IS(E_k, \theta) - IR(E_k, \theta)]/[IS(E_k, \theta) + IR(E_k, \theta)]$ (page 5). Apart from the fact that this is not what the arrows on top of Figure 2a indicate (they suggest that the XUV polarization is rotated), this requires changing the enantiomer in the COLTRIMS “regularly” (how often?) while keeping the exact same experimental conditions. How do the authors take into account possible artifacts coming from differences between the two enantiomers purity/enantiomeric excess? between different pressures of the two molecules in the COLTRIMS? Are the data shown in Figure 2a-b symmetrized and/or antisymmetrized? Comparing the colorbars in the RABBITT traces for the two enantiomers in Figure 3a-c (R MeOx) and Extended Data Figure 3 a-b (S MeOx), one clearly sees that the S enantiomer resulted in higher counts. The authors state that they have verified that the chiral asymmetry is also present when switching the XUV helicity in the same enantiomer, but no data is presented to support this claim. A measurement in a racemic mixture of MeOx, or with a linearly polarized XUV attosecond pulse train, could also have been shown to strengthen the validity of the conclusions.

We have exchanged the enantiomers in the COLTRIMS measurements every 6 hours and have run each measurement over 48 hours. In our definitions, the quantities $IR(E_k, \theta)$ and $IS(E_k, \theta)$ are photoelectron angular distributions normalized to their maximum. As a consequence of the chosen normalization scheme, absolute counts don't matter in our measurements (they are subject to varying laser parameters), only the relative counts between measurement sessions on R- vs. S-enantiomer matter. This approach eliminates signal variations that are not an intrinsically chiral signature of the studied molecules. For the same reason, the different total counts observed in Figure 3a-c (R MeOx) and Extended Data Figure 3 a-b (S MeOx) do not affect the results. In each case these measurements are self-referenced because only the phase shift between the forward- vs. backward-emitted photoelectrons is being extracted. The mirror symmetry visible in the results of Fig. 3F confirms the robustness of these results, as well as the chiral nature of the measured time delays.

We have used commercial samples, the specific batches of which were measured by the manufacturer to feature a >99.5% ee. This information has been added to the manuscript. This very high purity eliminates this source of artifacts.

Measurements with linearly polarized XUV could not be done under identical conditions because of technical reasons. However, the theta-resolved data shown in the manuscript already fulfills the request of the reviewer: the fact that the theta-resolved data shows mirror anti-symmetry with respect to $\theta=90^\circ$ demonstrates the chiral nature of the delays.

Third, the manuscript shows lacks of rigor in multiple points. It is unclear what is the harmonic spectrum used in the experiments. In Figure 1a, the spectrum shows harmonics H7 to H13, which is also written on page 3. But from Figure 2 on, the authors discuss SB14, suggesting interferences between H13 and H15. What is the actual spectrum used in the experiments?

We have added the measured high-harmonic spectrum in Fig. 1A. As expected, it contains H15. Although H15 is weak, it is sufficient to produce an oscillating signal at sideband 14.

It also looks like the authors do not use the same definition of the forward-backward time delay throughout the manuscript (forward minus backward or backward minus forward), making the presented results inconsistent with each other. Indeed, in Figure 3d SB8, the forward oscillation is ahead of the backward oscillation and the time delay in Figure 3f is positive. In Extended Data Figure 4 a and c (which corresponds to the same experiment), the forward electrons correspond to θ between $0-90^\circ$, and the backward electrons to θ between $90-180^\circ$. The forward electrons oscillations are here again ahead of the backwards

electrons (in agreement with Figure 3d), but in Extended Data Figure 4c, for R MeOx, if one calculates the time delay forward minus backwards it is now negative, in disagreement with Figure 3f.

We thank the referee for drawing our attention to this aspect. Actually our definitions are consistent with each other. Our definition of the forward-backward time delay (corresponding to Fig. 3f in the main text) is the amount of time the forward electron is delayed with respect to the backward electron. When the forward-backward time delay is positive, it means the forward electron is delayed with respect to the backward electron. For R-enantiomers, the forward electron is delayed, therefore the forward-backward time delay is positive shown in Fig. 3f in the main text. Now let us explain the definition of the y axis of Extended Data Fig. 4c and 4f (and in fact all angle-resolved time-delay figures). First it is not the XUV-IR delay. It is the opposite of the XUV-IR delay. When a SB oscillation peak appears at the larger XUV-IR delay, we call it delayed since in our experiments we physically delay the IR field. It means the ionization time delay is negative. This is why in Extended Data Fig. 4c, for R-enantiomers the ionization time delay of the forward electrons is negative and that of the backward electron is positive.

We have added the definition of the forward-backward time delay in the revised manuscript and have highlighted the meaning of the sign of the photoionization time delay in both main text and the caption of Figure 4.

Reply to reviewers' and editor's comments

We thank the reviewers and the editor for their reviews and thoughtful comments. Below, we reproduce **their statements in blue**, provide our replies in black and the **changes made to the manuscript in green**.

Reviewers' comments:

Referee #1 (Remarks to the Author):

The authors have added technical details on p.3 and p.11-12 of the manuscript highlighting the differences between their work and existing literature. As I wrote in my first review, they reside in the use of a COLTRIMS apparatus for the detection of the electrons in coincidence with ions and of an XUV attosecond pulse train for ionization. This is a different geometry of the detector and a different ionization regime compared to the literature. While this represents a challenging and elegant experiment, of interest to specialists, I do not think that it represents a conceptual advance for ultrafast photoionization of chiral molecules of interest to the broad readership of Nature.

We thank the reviewer for reiterating their qualification of our work as elegant and interesting.

We also appreciate the opportunity to further clarify the conceptual difference between our work and previous literature. As pointed out in our reply to the first report of the same reviewer, the conceptual advance of our work resides in its first application of circularly polarized attosecond pulses to chiral molecules, i.e. the demonstration of chiroptical spectroscopy with attosecond pulses.

This novel technique is used for the measurement of **Wigner delays** in the one-photon ionization of chiral molecules and the **coherent control of PECD**. As explained in our previous reply, neither of these two milestones have been achieved in previous work, simply because circularly polarized attosecond pulses are required for these two applications. References 18 and 21, to be discussed in more detail below, used >40-femtosecond pulses. They were moreover carried out in the non-perturbative regime of strong-field ionization, as opposed to the perturbative regime of RABBIT experiments in the present work. The conceptual innovations of our work can therefore be summarized as follows. Our work demonstrates:

- 1) How to measure Wigner delays in one-photon ionization of chiral molecules. Importantly, we show that these delays depend on both the polar (θ) and the azimuthal (ϕ) angle of photoemission in the light propagation direction. Therefore, three-dimensional momentum resolution of the photoelectrons is crucial for properly separating phase and amplitude modulations, which is demonstrated in the present work for the first time.
- 2) How to coherently control PECD. Importantly, and in contrast to previous work, this is the first experimental demonstration of coherent control over PECD. Previous works (e.g. Rozen et al., Ref. 21) demonstrated coherent control over other chiral asymmetries because they did not use circularly polarized light.

The fact that there is a delay between the electrons wavepackets emitted forwards or backwards in a chiral molecule, that is opposite for enantiomers, has been measured and calculated in Ref 18.

We thank the reviewer for raising this important point. While Ref. 18 (Beaulieu et al., *Science* 2017) indeed reported phase shifts in photoelectron emission that differ between forward and backward directions and between enantiomers, we respectfully highlight several critical

differences between that work and the present study, both in conceptual scope and methodological rigor:

1. **Measurement Regime and Interpretation**

The phase shifts reported in Ref. 18 were obtained using two-color strong-field ionization in the non-perturbative regime, where many multiphoton ionization pathways contribute to the final photoelectron signal. In this regime, the measured “delays” are strongly dependent on laser intensity (as demonstrated in Ref. 30) and lack a clear theoretical link to intrinsic molecular properties such as the Wigner delay. In contrast, our work uses a perturbative RABBITT scheme with circularly polarized attosecond XUV pulses and weak IR dressing, which enables quantitative extraction of one-photon-ionization time delays that are fundamentally tied to molecular properties.

2. **Angular Resolution and Dimensionality**

Ref. 18 employed velocity-map imaging (VMI), which captures a 2D projection of the photoelectron momentum distribution. This method does not resolve the azimuthal angle (ϕ) of emission, which—as shown in our work—is essential to disentangle amplitude and phase contributions in the presence of circular polarization. The 3D momentum resolution of our COLTRIMS measurements is therefore crucial to isolate the true photoionization time delays and to map their dependence on both polar (θ) and azimuthal (ϕ) angles.

3. **State Resolution via Coincidence Detection**

In Ref. 18, the photoelectron spectrum was recorded without coincidence detection. As a result, multiple ionic states contributed to the observed signal, leading to spectral congestion and ambiguity in identifying the ionization pathways. In our work, electron-ion coincidence detection enables unambiguous assignment of photoelectrons to specific molecular ionic states, which is critical for extracting meaningful time delays.

4. **Quantitative Theoretical Support**

To date, the phase shifts observed in Ref. 18 have not been quantitatively reproduced by theory, likely due to the complexity of strong-field dynamics. By contrast, the perturbative framework of our experiment enables direct comparison to ab initio simulations, showing good agreement and reinforcing the validity and quantitative interpretation of our delay measurements.

5. **Conceptual Advancement**

Our work introduces attosecond chiroptical spectroscopy using circularly polarized XUV pulses, enabling for the first time the quantitative measurement of chiral photoionization delays resolved, notably resolved in all emission angles in the light propagation frame. This fundamentally advances the field from a qualitative to a quantitative discipline, as pointed out by reviewer 2, and enables novel insights into the temporal dynamics of chiral photoemission.

In summary, while Ref. 18 made important pioneering observations, our work builds on and surpasses those results by providing a quantitative, angle- and state-resolved measurement of chiral photoionization delays in the perturbative regime, backed by rigorous theoretical interpretation and coincidence-resolved detection. These advances are essential for establishing a quantitative framework for attosecond-resolved chiral dynamics.

The fact that a chiral signal in the photoionization of a chiral molecule can be controlled with light on the subcycle timescale has been shown experimentally and theoretically in Ref 29.

We thank the reviewer for highlighting this important prior contribution. Ref. 29 (Rozen et al., *Phys. Rev. X* 2019, now ref. 21) indeed demonstrated subcycle control over chiral-sensitive photoelectron asymmetries using intense two-color femtosecond laser fields. However, it is important to underscore the fundamental differences between that work and the present study in terms of both observables, light fields, and the physical mechanisms underlying the control:

1. **Different Observables**

While Ref. 29 achieved coherent control over chiral forward-backward asymmetries, these asymmetries are not equivalent to photoelectron circular dichroism (PECD). PECD is defined as the forward-backward asymmetry in the photoelectron distribution induced by circularly polarized light, and it remains nonzero after hemispherical integration. In contrast, the asymmetries in Ref. 29 were induced by linearly polarized fields and vanish upon hemispherical integration. Thus, the physical observable studied in Ref. 29 is distinct from PECD and belongs to a different class of chiral effects.

2. **Ionization Regime and Mechanism**

Ref. 29 operated in the non-perturbative strong-field ionization regime, involving multiple competing multiphoton pathways. This results in complex dynamics that are difficult to interpret and model quantitatively. In contrast, our work is conducted in the perturbative RABBIT regime, where only two well-defined quantum pathways contribute to each final photoelectron energy. This allows for a transparent and mechanistically clear interpretation of coherent control in terms of interference between partial waves, enabling the first realization of coherent control over PECD itself.

3. **Attosecond Control with Circularly Polarized XUV Pulses**

Our study is the first to demonstrate attosecond control of PECD using circularly polarized attosecond XUV pulses. This is a substantial technical and conceptual advance, as it allows the coherent manipulation of true chiral observables (PECD) with subcycle precision in a regime that supports a direct link to fundamental molecular properties. The first use of circularly polarized attosecond pulses moreover opens the door to quantitative attosecond chiral spectroscopy.

In summary, while Ref. 29 (now ref. 21) provided valuable insights into the control of laser-induced chiral asymmetries using strong-field techniques, our work constitutes the **first attosecond-resolved coherent control of PECD**, based on a well-understood, perturbative mechanism involving circularly polarized light and angle-resolved detection. This marks a **new conceptual and methodological milestone** in the study of chiral photoionization dynamics.

The fact that the PECD values can differ in sign and magnitude (be “enhanced”) depending on the ionization scheme (single photon XUV, multiphoton IR, multiphoton visible... two-photon XUV + IR in the present manuscript) has been thoroughly shown in the literature (e.g. *J. Chem. Phys.* 139, 234307 (2013) in Camphor).

We thank the reviewer for highlighting this well-established aspect of PECD. Prior studies, including the cited work (*J. Chem. Phys.* 139, 234307 (2013)), have indeed shown that PECD can vary significantly in both sign and magnitude depending on the ionization scheme employed—whether single-photon XUV, multiphoton IR or visible, or two-color excitation schemes. These variations arise from differences in the contributing partial-wave interferences and intermediate states involved in the ionization process.

The key conceptual advances of the present work go beyond this established scheme dependence. First, we demonstrate **coherent control over PECD on attosecond**

timescales using a well-defined perturbative two-photon ionization scheme involving circularly polarized attosecond XUV pulses and a weak IR dressing field. This allows for a **precise mechanistic understanding** of PECD modulation based on the interference of only two quantum pathways, unlike in strong-field regimes where interpretation is complicated by many competing channels.

Second, we show that the **polarization state of the IR field**—linear, co-rotating circular, or counter-rotating circular with respect to the XUV field—has a **qualitative effect on the PECD**, further enhancing the observed asymmetry by various amounts. These findings establish a new level of control over chiral photoionization dynamics.

Together, these advances merge the fields of attosecond spectroscopy and coherent control, offering a new and quantitative approach to exploring the ultrafast dynamics of chiral molecules, which can directly be transferred to the X-ray regime accessible with free electron lasers.

The assessment of the calculations that are compared to the experimental results is out of my field of expertise. As in the previous submission, they still rely heavily on Ref 4 which is under review at another journal according to the authors.

Although Ref. 4 is not published yet, the computational methods used in Ref. 4 have already been described and validated in Refs. 3 and 29 that have been published in PRL and JCP, respectively, in 2019.

The authors have clarified technical points that I raised, but they were not the main reason why I did not recommend publication of this manuscript in Nature. The main reason being the lack of fundamental new insight regarding ultrafast photionization of chiral molecules as detailed above, which still holds in the revised version. Therefore I do not recommend publication of this manuscript in Nature.

We thank the reviewer for their time and efforts spent on the review of this manuscript. We hope that this revision has further clarified the fundamental conceptual advances that our work represents over the previous literature.

Referee #2 (Remarks to the Author):

The work of Meng Han et al. presented in "Attosecond Coherent Control and Measurement of the Photoionization Dynamics of Chiral Molecules" constitutes a leap forward in our capabilities to study attosecond chiral molecular dynamics. It must be said that this is not the first work on the subject, and as rightfully pointed out by another referee, one must properly acknowledge these pioneering works by Beaulieu et al. presented in *Science* (2017) and by Rozen et al. in *Physical Review X* (2019). Over the past years and decades, different strategies have been implemented to study how chiral effects depend on both structure and dynamics in molecules. The present work is elegant and it is the first "RABBIT" experiment with circularly polarized attosecond pulses aimed to resolve chiral properties of molecules in space and time. It shows that it is possible to enhance and coherently control chiral properties. Such control has been shown previously using other techniques. This means that we must ask ourselves if the adaptation of the "RABBIT" technique to this type of problem warrants publication in *NATURE*. I believe that it does because the RABBIT scheme will help to transform this research field from a phenomenological field of laser-induced dynamics to a quantitative and systematic research field for studies of chirality of molecules in space and time.

We sincerely thank the referee for their thorough and thoughtful evaluation of our work and for recognizing the significance of our contribution to the field of attosecond dynamics of chiral molecules. We greatly appreciate the referee's careful distinction regarding the prior pioneering works by Beaulieu et al. (*Science* 2017) and Rozen et al. (*Physical Review X* 2019), and we fully agree that these important studies deserve additional acknowledgment. **In the revised manuscript, we have explicitly mentioned these works early on (on p. 2). A detailed discussion of the fundamental differences between the present work and those works is given on p. 11.**

We are grateful for the referee's positive assessment of the novelty and potential impact of our development of a chiral-sensitive RABBIT and coherent-control technique. We agree with the referee's view that our approach defines a more quantitative and systematic exploration of chiral phenomena on attosecond timescales, and we are encouraged by the referee's endorsement that our work merits publication in *Nature*.

Prior works attaining attosecond delays experimentally in chiral molecules are criticized by the authors, as lacking a proper theoretical basis for interpreting the experiments. There is a fine line here, and while I see the great value in these prior works, I believe that the authors are correct in their assessment that the RABBIT technique will open up for quantitative studies of molecular chiral dynamics that go beyond qualitative laser-driven observations. The strength of RABBIT, which has previously been used for attosecond pulse characterization and for understanding photoionization delays and correlation effects in atoms, is that it provides a platform to perform quantitative experiments that produce numbers that can be compared with numbers from theoretical models with different approximations. In this way, we can test models and gain a better understanding of the system under study.

We fully agree with the reviewer on the fundamental importance of these aspects. It is clear that the pioneering works based on intense femtosecond pulses as reported in Refs. 18 and 29 (now 21) demonstrated important principles of strong-field-ionization dynamics of chiral molecules, but as correctly pointed out by the reviewers, these results could not be interpreted quantitatively. The phase shifts measured in two-color strong-field ionization (Ref. 18) are known to depend on the intensity of the laser fields, such that they cannot be related to fundamental molecular properties, such as Wigner delays. Neither reference 18, nor any subsequent publication, have reported comparisons of the results of Ref. 18 with theory. A similar situation applies to the strong-field control over the asymmetry of photoemission

shown in Ref. 21 (which is not PECD). Although TDSE simulations have been reported in Ref. 21, they were limited to a “toy model” (in the language of Ref. 21), instead of the actual chiral molecule, such that they could obviously not reproduce the experimental observations. No quantitative interpretation of the data of Ref. 21 has been published over the past 6 years. As argued in our first response to reviewer 1, the interpretation of strong-field control is very difficult because many multiphoton pathways contribute to a given final photoelectron kinetic energy. In contrast, both delay measurements and coherent control in the perturbative “RABBIT” regime isolate pairs of quantum paths, which give rise to a transparent and unique theoretical interpretation in the framework of second-order perturbation theory, which relates the experimental observables to fundamental, intrinsic molecular quantities (Wigner delays, continuum-continuum delays, etc.) on one hand and a transparent interpretation of coherent control over PECD (in terms of interference of pairs of partial waves) on the other hand. As kindly pointed out by the reviewer, our work thus marks the beginning of quantitative research on the photoionization dynamics of chiral molecules and the coherent control of chiral photoionization, which hold the promise of becoming two active branches of attosecond spectroscopy.

The experimental results are interpreted using theoretical molecular simulations including continuum-continuum dynamics. The agreement between experiment and theory is convincing with similar trends observed in time delays over energy and electron-emission angle. Thus, the frozen core approximation seems adequate for the study. In summary, the usage of the RABBIT technique is the major selling point of the work, as it implies measurements of “intrinsic properties” of the chiral molecules in space and time.

We thank the reviewer for their appreciative comments.

Referee #3 (Remarks to the Author):

In the manuscript entitled “Attosecond coherent control and measurement of the photoionization dynamics of chiral molecules”, the authors of Meng Han et al. reports measurements of attosecond time-resolved photoelectron spectrograms of the two enantiomers of the chiral molecule methyloxirane. Technically, as far as I know, this is the first time that circularly polarized attosecond extreme ultraviolet pulses have been used to explore ultrafast photoelectron dynamics. In addition, the three-dimensional momentum distribution of photoelectrons has been measured using COLTRIMS technology. Their experimental results not only achieved significantly enhanced photoelectronic circular dichroism (PECD) effect, but also enabled them to distinguish and understand PECD from more perspectives, such as electronic states, angles, energies, and enantiomers.

We thank the reviewer for their thoughtful and accurate summary of our work.

Due to two-color two-photon ionization located in the perturbation regime, the time delay of photoemission measured by RABBIT type has a well-known and clear physical concept, namely the Wigner delay, which is associated with the inherent properties of the target molecule. However, two-color strong-field ionization locates in non-perturbative regime, involving multiphoton ionization and even tunneling ionization, with more complex photoionization dynamics and thus it is difficult to quantitatively explain the underlying mechanisms. There is no physical concept corresponding to the photoemission time delay measured based on this technology that can be understood. Therefore, the current work provides more complete theoretical supports for the attosecond measurement and control of chiral molecules.

We are grateful to the reviewer for pointing out the important fundamental differences between measuring phase shifts in strong-field ionization and time delays in RABBIT experiments.

The current work enables the direct manipulation of photoelectron circular dichroism with attosecond - level precision, representing a significant advancement compared to previously published works (in contrast to femtosecond - scale precision). I believe that the current work represents a significant advancement in attosecond metrology using circularly polarized pulses, enabling attosecond - scale coherent control and measurement of the photoionization dynamics of chiral molecules.

We thank the reviewer for emphasizing the important advance represented by our first application of circularly polarized attosecond pulses to chiral molecules. All previous experiments have indeed relied on femtosecond pulses and it is only with the availability of characterized circularly polarized attosecond pulses that the present achievements have come within reach.

However, compared to Reference 4, aside from the experiments, there are no new conceptual discoveries or breakthroughs. If the current work can overcome this lack of conceptual innovation, it should be able to attract the interest of a wide range of researchers in the fields of attosecond science and chiral molecule research. In that case, I would be happy to recommend its publication in Nature.

We sincerely thank the referee for their thoughtful engagement and for recognizing the technical achievements of our work. We greatly appreciate the constructive suggestion that articulating a clear conceptual advance would significantly strengthen the manuscript's impact and relevance to the broader attosecond and chiral molecule communities.

In response, we would like to clarify and highlight the conceptual innovations that distinguish the present study from our prior theoretical work (Ref. 4). While Ref. 4 focused exclusively on the **theoretical proposal of coherent control of PECD**, it did not address or consider **photoionization time delays**, nor did it present an experimental realization of the proposed control mechanisms.

In the present manuscript, we demonstrate for the first time a **quantitative and angle-resolved measurement of chiral photoionization time delays** using circularly polarized attosecond XUV pulses and a weak IR probe. This marks a **conceptual breakthrough**: it establishes a new measurement protocol, relying on the present demonstration of **chiroptical attosecond chronoscopy**, which provides the first experimental access to intrinsic chiral molecular properties known as **Wigner delays**, additionally with full angular (θ , φ) resolution in the light-propagation frame. Such delays, and their dependence on emission angles in the laboratory frame, were neither considered nor discussed in Ref. 4.

Furthermore, we have experimentally realized the mechanism of **coherent control of PECD through interference between well-defined partial waves** in the perturbative regime. While the concept of pathway interference is a brilliant concept that has driven research in the coherent-control community for 40 years, its implementation in **chiral molecules** has only become possible through the generation and characterization of **circularly polarized attosecond pulses**, a very recent technical advance of our group. The merging of these two fields—**attosecond chiroptical spectroscopy** and **coherent control**—in a single experiment opens entirely new avenues for controlling and understanding chiral phenomena on their natural timescale, which can directly be transferred into the X-ray regime and thereby opens new research perspectives for free-electron lasers.

Finally, we emphasize that resolving both **polar (θ)** and **azimuthal (φ)** emission angles is essential for extracting unambiguous chiral photoionization delays. The ability to perform this analysis, thanks to our use of full 3D electron-ion coincidence detection, represents a fundamental step forward. This angular resolution not only enables the **disentangling of phase and amplitude effects** in chiral photoemission, but it is also **necessary for a correct interpretation** of the measured delays—something that, to our knowledge, has never been achieved before.

We hope that this clarification helps to convey the distinct and complementary conceptual contributions of the present work, and that the referee may view this integration of attosecond chronoscopy, angularly resolved detection, and coherent control as a compelling conceptual advance that meaningfully extends beyond the scope of Ref. 4.

We are grateful once again for the referee's valuable comments and hope that this revised presentation addresses the concerns raised.

We have highlighted these conceptual innovations at several locations in the manuscript:

Abstract:

"We demonstrate that chiral photoionization delays depend on both polar and azimuthal angles of photoemission in the light-propagation frame, requiring three-dimensional momentum resolution. We measure forward-backward chiral-sensitive delays of up to 120 as and polar-angle-resolved photoionization delays up to 240 as"

p. 2:

"It also reveals the characteristic dependence of chiral photoionization delays on both the polar and the azimuthal angles of photoemission in the light-propagation frame."

p. 9/10:

“It is important to realize that 3D momentum resolution is a prerequisite for a quantitative measurement of chiral photoionization delays because the latter depend on both laboratory-frame angles of photoemission (θ and φ) in any chiral-sensitive experiment. We illustrate this fact by showing in Extended Data Fig. 4 how the phase of SB8 depends on φ in the case of co-rotating XUV and IR fields. We find that the phase linearly depends on the angle φ , which is the RABBIT analogue of the angular streaking principle. Taking this property into account, allows us to perform a quantitative analysis of angle-resolved chiral photoionization delays.”

p. 21:

“For co- or counter-rotating XUV and IR pulses, this pathway interference gives rise to the terms with $M = \pm 2$, which oscillate as function of the XUV-IR delay with frequency $2\omega_{\text{IR}}$, where ω_{IR} is the frequency of the IR pulse. The $M = 0$ terms only contain the single-pathway contributions and do not oscillate with the delay. In contrast, an IR pulse linearly polarized in the polarization plane of the XUV pulse gives rise to both interfering and non-interfering contributions for both $M = 0$ and $|M| = 2$. These, however, differ in their delay dependence: for $M = 0$, the interference terms oscillate with $2\omega_{\text{IR}}$ whereas the single-pathway contributions do not oscillate. For $|M| = 2$, the situation is reversed. In the angle-resolved PECD, $M = 0$ terms are maximal at $\theta_k = 0^\circ$ and 180° , whereas terms with $|M| = 2$ are maximal at $\theta_k \approx 55^\circ$ and 125° . Changing IR polarization and delay thus enables control over the anisotropy in the angle-resolved PECD. Furthermore, averaging over the delay spatially separates the single-pathway and interference contributions on the detector plane, as illustrated in Extended Data Figure X. This may be utilized to selectively extract single-pathway and interference contributions from delay-resolved and delay-averaged PECD spectra.”

Additionally, we have added the new Extended Data Fig. 4 and its discussion to illustrate the importance of the phi-resolved analysis of the photoionization delays.